# FROZEN PRIORS, FLUID FORECASTS: PREQUENTIAL UNCERTAINTY FOR LOW-DATA DEPLOYMENT WITH PRETRAINED GENERATIVE MODELS

**Fernando Ruiz-Mazo**
Department of Computer Science, Aalto University;
DPMMS, University of Cambridge
fr440@cam.ac.uk

**Vikas Garg**
Aalto University
YaiYai Ltd
vgarg@csail.mit.edu

## ABSTRACT

Deploying ML systems with only a few real samples makes operational metrics (such as alert rates or mean scores) highly unstable. Existing uncertainty quantification (UQ) methods fail here: frequentist intervals ignore the deployed predictive rule, Bayesian posteriors assume continual refitting, and conformal methods offer per-example rather than long-run guarantees. We introduce a forecast-first UQ framework that blends the empirical distribution with a frozen pretrained generator using a unique Dirichlet schedule, ensuring time-consistent forecasts. Uncertainty is quantified via martingale posteriors: a lightweight, likelihood-free resampling method that simulates future forecasts under the deployed rule, yielding sharp, well-calibrated intervals for both current and long-run metrics without retraining or density evaluation. A single hyperparameter, set by a small-$n$ minimax criterion, balances sampling variance and model–data mismatch; for bounded scores, we provide finite-time drift guarantees. We also show how this framework informs optimal retraining decisions. Applicable off-the-shelf to frozen generators (flows, diffusion, autoregressive models, GANs) and linear metrics (means, tails, NLL), it outperforms bootstrap baselines across vision and language benchmarks (WikiText-2, CIFAR-10, and SVHN datasets); e.g., it achieves $\sim$90% coverage on GPT-2 with 20 samples vs. 37% for bootstrap. Importantly, our uncertainty estimates are operational under the deployed forecasting rule agnostic of the population parameters, affording practicable estimators for deployment in real world settings. Code available at https://github.com/Aalto-QuML/Prequential/.

## 1 INTRODUCTION

When a machine learning system is deployed, it must begin operating on *real-world data*, often with only a handful of examples available. Despite this data scarcity, practitioners still need to answer key operational questions early on regarding how the model will behave in the long run. For example:

- *What fraction of cases will trigger an alert if we keep using the system?* (e.g., for an automated trading setup, the proportion of minutes that would hit a safety stop)
- *What is the expected value of a particular score or metric in the long term?* (e.g., for a new health program or app, the long-run average outcome per person, such as "high blood pressure".)

Formally, let $Y \in \mathcal{Y}$ be a future outcome and $h : \mathcal{Y} \to \mathbb{R}$ a fixed score function. Ideally, to respond to these questions, we would calculate the expected value of the score function under the true (but unknown) data distribution $Y \sim F^\star$:

$$\theta(F^\star) := \int h \, dF^\star.$$

This includes quantities like $\Pr\{h(Y) > \tau\}$ (a tail) and $\mathbb{E}[h(Y)]$ (a mean). However, with only a small sample $Y_{1:i}$, the empirical distribution $\widehat{F}_i$ is unstable and yields unreliable estimates.

Throughout, let $n$ denote the number of *real* observations actually seen in the new deployment environment. In staged rollouts and safety-gated launches, $n$ is intentionally small at the beginning due to human quality checks, rate limiting, and policy restrictions. Our methods are expressly designed for this small-$n$ regime, where classical resampling often undercovers.

A practical workaround (used in domains such as translation, vision, and medical imaging (Sennrich et al., 2016; Azizi et al., 2023; Yi et al., 2019) is to borrow stability from a pretrained generative model trained on similar data (see section A in the Appendix for more details). Let $Q_\phi$ be the distribution defined by such a model. Importantly, we do *not* assume $Q_\phi = F^\star$; instead, we use it as a prior-like stabilizer that fades out as more real data arrives.

## 1.1 CONTRIBUTIONS

| Summary of our contributions | |
|---|---|
| **New forecasting method:** | |
| Prequential Blend | Section 3 |
| **Theoretical Analysis:** | |
| Adjusting the blend | Section 4 |
| Adjusting the predictive horizon | Section 6 |
| Main algorithm (Martingale Posterior adaptation) | Section 5 |
| **A use case of the framework:** | |
| Optimal retraining of deployed generative models | Section 7 |
| **Experiments:** | |
| Two Moons C.1, GPT-2 C.2, CIFAR-10 (ID C.3.1) and SVHN (OOD C.3.2) | Section 8 |

Operationally, our target is the long-run metric under the deployed rule, $\theta_\infty$, rather than the population parameter $\theta(F^\star)$.

Given this setting, how do we integrate the "synthetic" data generated by this generative model $Q_\phi$ with the data we start receiving from the real distribution $F^\star$? How do we do it in a way that is mathematically sound and rigorous? And even more, are there multiple ways of doing this or just one? To answer these questions, we propose a *prequential forecasting approach* (Dawid, 1984; Fortini & Petrone, 2025), where at each time step $i$ (each new data point/batch arrives) we blend the empirical data distribution with the fixed generator one:

$$P_i = (1 - \lambda_i)\,\widehat{F}_i + \lambda_i\,Q_\phi, \quad \text{with} \quad \lambda_i = \frac{\alpha}{i + \alpha}, \ \alpha > 0.$$

showing in section 3 that this Dirichlet-style blending rule is the *only affine combination* that ensures *time-consistent forecasts*: for any bounded score $h$, the sequence $\{\theta(P_i)\}$ forms a martingale. This means forecasts are stable early on (shrinking toward the model) and gradually revert to empirical behavior as data accumulates.

> **Reality vs. Surrogate Forecasting**
>
> **We do not claim that real data follows our forecasts.** The true data distribution $F^\star$ may differ from both the generator and the blended forecast $P_i$. We deliberately choose this *surrogate forecasting system* to stabilize small-sample metrics. Our theoretical results (on coherence, martingales, and uncertainty quantification) apply to this surrogate process and its operational targets $\theta(P_i)$, not to $\theta(F^\star)$.

Crucially, this same mechanism enables *uncertainty quantification* (UQ) without retraining or likelihoods. By simulating future forecasts under the deployed rule and tracking only the score values, we construct in section 5 a *martingale posterior (MP)* (Fong et al., 2023), a calibrated predictive distribution over the long-run metric $\theta_\infty$. This yields lightweight, coherent uncertainty intervals that reflect what operations actually act on: the forecasted score under the deployed blend.

Furthermore, we also give a principled way of choosing the hyperparameter $\alpha$ in our blend (section 4) as well as the predictive horizon to simulate our MP resamples (section 6). Finally, we leverage our theory to analyze how our framework may be used to optimally decide when to retrain a deployed generative model.

**Why not employ standard Uncertainty Quantification (UQ) techniques?**    Frequentist intervals target population parameters of the unknown data law and ignore the *deployed* predictive rule, often becoming conservative at small $n$. Bayesian posteriors integrate *parameter* uncertainty and typically presume continual refitting, which conflicts with frozen deployments. Conformal prediction provides per-example coverage rather than uncertainty for *long-run rates* under a fixed rule (Vovk et al., 2005; Angelopoulos & Bates, 2023). Raw model likelihood is unreliable under shift (e.g., flows giving higher likelihood to out-of-distribution (OOD) samples than in-distribution (ID) ones), and likelihood-ratio fixes still rely on density modeling (Nalisnick et al., 2019; Ren et al., 2019; Kirichenko et al., 2020). A more in-depth discussion of these techniques can be found in section A of the Appendix. These limitations of the standard methodologies motivate our approach.

**Scope.**    The method applies to any *frozen* generator with a fixed sampler and evaluable score $h$: explicit-density models (flows, VAEs; with $h(y) = -\log q_\phi(y)$ when available), diffusion/score-based models with fixed schedules, autoregressive LMs with fixed decoding, implicit samplers (GANs), and EBMs with a frozen MCMC kernel. **Not for:** online updates to parameters/decoding/M-CMC/prompts (breaks coherence); targeting population parameters of $F^\star$ (prefer frequentist tools); or per-example coverage (prefer conformal prediction (Angelopoulos & Bates, 2023)). We assume a *frozen* generator $Q_\phi$: the parameters $\phi$ are fixed at deployment start, so $Q_\phi$ is $\mathcal{F}_i$-measurable for all $i$; any internal sampling randomness (e.g., diffusion noise or a GAN latent $z$) is taken conditional on $Q_\phi$ and is not updated online. Because we only require sampling $X \sim Q_\phi$ and evaluating $h(X)$, the procedure is architecture-agnostic and applies under both domain and semantic shifts.

## 2    PRELIMINARIES

**At a glance.**    We set up the prequential (forecast-first) framework and notation. Under (scalar) prequential coherence, the one-step forecasts $(P_i)$ make the scalar functional $\theta(P_i) = \int h \, dP_i$ a martingale for any bounded (or $L^2$) score $h$, hence $\theta(P_i) \to \theta_\infty$. This justifies targeting the long-run operational metric $\theta_\infty$ and using the *martingale posterior (MP)* $\Pi_{\mathrm{MP}}(\cdot \mid \mathcal{F}_n)$ for uncertainty; a Freedman-type deviation bound later sizes simulation horizons and provides a conservative drift diagnostic.

This section recalls the *prequential* (forecast-first) viewpoint and the minimal martingale machinery we use later. Mainly, this point of view of statistics puts our source of uncertainty in the outcomes we have not seen yet $(Y_{n+1}, Y_{n+2}, ...)$ contrary to the Bayesian approach, which puts it on the parameters of our distribution; see Dawid (1984) and surveys such as Fortini & Petrone (2025) for background. For martingale posteriors see Fong et al. (2023).

**Data stream and forecasts.**    Let $(\mathsf{Y}, \mathcal{B})$ be a measurable outcome space and $Y_1, Y_2, \ldots$ the observed sequence with $\mathcal{F}_i = \sigma(Y_{1:i})$. A *forecasting system* is an $\mathcal{F}_i$–adapted sequence of probability measures

$$P_i : \ \Omega \to \mathcal{P}(\mathsf{Y}), \qquad i \geq 0,$$

meaning that for every $A \in \mathcal{B}$ the map $\omega \mapsto P_i(\omega)(A)$ is $\mathcal{F}_i$–measurable. Equivalently, $(P_i)$ is a stochastic kernel from $(\Omega, \mathcal{F}_i)$ to $(\mathsf{Y}, \mathcal{B})$. At time $i{+}1$ $P_i$ is computed and then $Y_{i+1}$ is realized; under ideal calibration one has

$$P_i(\cdot) \ = \ \Pr\big(Y_{i+1} \in \cdot \,\big|\, \mathcal{F}_i\big).$$

**Prequential calibration (scalar coherence).**    A one-step forecasting system $(P_i)$ is (scalar) *coherent* for a class $\mathcal{H}$ of test functions if, for every $h \in \mathcal{H}$,

$$\mathbb{E}\big[h(Y_{i+1}) \,\big|\, \mathcal{F}_i\big] \ = \ \int h \, dP_i \qquad (i \geq 0).$$

We will take $\mathcal{H}$ to contain at least all *integrable* $h$.

**Linear functionals and martingales.** Fix a (bounded or square-integrable) score $h$. Define the linear functional $\theta(F) = \int h \, dF$ and the process

$$\theta(P_i) \;=\; \int h \, dP_i \qquad (i \geq 0).$$

Under calibration, $\mathbb{E}[\theta(P_{i+1}) \mid \mathcal{F}_i] = \theta(P_i)$, so $(\theta(P_i))$ is a martingale. If $h$ is bounded, or $\sup_i \mathbb{E} \int h^2 \, dP_i < \infty$, Doob's martingale convergence theorem yields

$$\theta(P_i) \;\longrightarrow\; \theta_\infty \quad \text{almost surely and in } L^2,$$

where $\theta_\infty$ is the *long-run value* of the operational metric under the given forecasting system (Doob, 1953).

**Martingale posterior (MP).** Following Fong et al. (2023), the *martingale posterior* for the target is the conditional law of the limit:

$$\Pi_{\mathrm{MP}}(\cdot \mid \mathcal{F}_n) \;:=\; \mathrm{Law}\big(\theta_\infty \mid \mathcal{F}_n\big).$$

Operationally, one approximates $\Pi_{\mathrm{MP}}$ by *predictive resampling 1*: starting from the observed history and the forecasting system $(P_i)$, simulate future one-step forecasts and outcomes, track the functional $\theta(P_i)$ forward, and use long-horizon values as draws from the MP. This keeps inference forecast-centric and likelihood-agnostic.

**Deviation bounds.** When $h$ is bounded, the increments $\Delta_{i+1} := \theta(P_{i+1}) - \theta(P_i)$ are bounded, and their conditional variances accumulate as $V_t := \sum_{i=n}^{t-1} \mathbb{E}[\Delta_{i+1}^2 \mid \mathcal{F}_i]$. Freedman's inequality gives, for any $\delta \in (0, 1)$ and all $t \geq n$,

$$\big| \theta(P_t) - \theta(P_n) \big| \;\leq\; \sqrt{2 \, V_t \log(2/\delta)} \;+\; \tfrac{c}{3} \log(2/\delta) \quad \text{with probability at least } 1 - \delta,$$

where $c$ bounds $|\Delta_{i+1}|$ almost surely (see Freedman, 1975). This provides a generic, prequential "drift" control between the current forecasted functional and future values.

**What to keep in mind.** (i) Prequential analysis is *forecast-first*: objects of interest are the issued $P_i$'s and functionals thereof, not parameter posteriors. (ii) Calibration implies martingale structure for $\theta(P_i)$, enabling convergence and uncertainty quantification via MPs. (iii) Concentration tools for martingales provide finite-time, deviation controls that we will use to size simulation horizons and report conservative drift terms.

## 3 The Predictive Rule

**At a glance.** Among affine empirical–model blends with predictable weights, requiring forecasts that "do not move on average" (scalar prequential coherence) *forces* the Dirichlet schedule $\lambda_i = \alpha/(i + \alpha)$. This makes $\theta(P_i) = \int h \, dP_i$ a martingale for bounded (or $L^2$) scores $h$, so $\theta(P_i) \to \theta_\infty$. We will later use this to size MP horizons (via deviation bounds) and to connect the conditional mean $\mathbb{E}[\theta_\infty \mid \mathcal{F}_n]$ to the familiar shrinkage point.

As stated, we study a simple, transparent prequential rule for the *deployed* forecasts $P_i$:

$$P_i \;=\; (1 - \lambda_i) \, \widehat{F}_i \;+\; \lambda_i \, Q_\phi, \qquad \widehat{F}_i = \tfrac{1}{i} \sum_{k=1}^{i} \delta_{Y_k},$$

with predictable weights $\lambda_i \in [0, 1]$ (measurable w.r.t. $\mathcal{F}_{i-1}$). The data are generated prequentially: $\Pr(Y_i \in A \mid \mathcal{F}_{i-1}) = P_{i-1}(A)$ (Dawid, 1984).

> **Intuition: forecasts that don't move on average**
>
> In the prequential view, today's forecast should be the best predictor of tomorrow's forecast. Testing only scalar quantities, this says that for every bounded score $h$,
>
> $$\mathbb{E}\left[\int h\,dP_i \,\middle|\, \mathcal{F}_{i-1}\right] = \int h\,dP_{i-1}.$$
>
> Within affine blends with predictable weights, this *forces* the Dirichlet/Pólya pseudo-count schedule $\lambda_i = \alpha/(i+\alpha)$ (Blackwell & MacQueen, 1973; Ferguson, 1973); early forecasts borrow from $Q_\phi$ and then fade to the empirical law.

**Theorem 1** (Coherence $\Leftrightarrow$ Dirichlet; scalar consequences). *Fix $n \in \mathbb{N}$, $\mathcal{F}_i = \sigma(Y_{1:i})$, and assume $Q_\phi$ is $\mathcal{F}_n$-measurable (frozen post-training). For $i \geq n$ let $P_i = (1-\lambda_i)\widehat{F}_i + \lambda_i Q_\phi$ with $\lambda_i$ predictable, and suppose the prequential law holds for $i \geq n+1$.*

*Then the following are equivalent:*

*(A)* Scalar coherence for all bounded $h$: $\mathbb{E}\big[\int h\,dP_i \mid \mathcal{F}_{i-1}\big] = \int h\,dP_{i-1}$ *for all* $i \geq n+1$.

*(B)* Dirichlet weights: $\lambda_i = \alpha/(i+\alpha)$ *for some* $\alpha > 0$ *and all* $i \geq n$.

*Under either condition, for any bounded (or $L^2$) score $h$ the process $\theta(P_i) := \int h\,dP_i$ is a martingale and*

$$\theta(P_i) \longrightarrow \theta_\infty \quad \text{a.s. and in } L^2, \qquad \mathbb{E}[\theta_\infty \mid \mathcal{F}_n] = \frac{n}{n+\alpha}\int h\,d\widehat{F}_n + \frac{\alpha}{n+\alpha}\int h\,dQ_\phi.$$

**Why this matters.** **(i) Scoped uniqueness.** If you want an interpretable empirical–model blend with predictable weights *and* you want forecasts that "don't move on average," you must use $\lambda_i = \alpha/(i+\alpha)$. **(ii) Direct operational payoff.** For any fixed score $h$, $\theta(P_i) = \int h\,dP_i$ is the quantity operations care about (mean rate, tail mass, NLL). The limit $\theta_\infty$ is its long-run value, and its conditional mean equals the familiar shrinkage point. Convergence and the mean identity follow from standard martingale theory (Doob, 1953).

**Practical tie-in (MP and resampling).** We quantify uncertainty for $\theta_\infty$ via the *martingale posterior* (the conditional law of $\theta_\infty$ given $\mathcal{F}_n$) using predictive resampling in the spirit of Fong et al. (2023): simulate forward under the rule $\lambda_i = \alpha/(i+\alpha)$ with $P_i = (1-\lambda_i)\widehat{F}_i + \lambda_i Q_\phi$, track only running values of $h$, and use long-horizon values as MP draws (see Algorithm 1). For a conservative stopping rule for the simulation horizon, see Section 6; for a principled retraining trigger, see Section 7.

**What breaks if we violate scope.** *Fixed $\lambda$* (no decay) or *non-predictable* weights violate scalar coherence on generic histories; the martingale property fails. Updating $\phi$ online breaks $\mathcal{F}_n$-measurability of $Q_\phi$ and the coefficient recursion that yields the Dirichlet schedule.

## 4 How to choose $\alpha$ optimally

We want a single, transparent way to pick the knob $\alpha$ that sets the coherent shrinkage weight $\lambda = \alpha/(n+\alpha)$ in the estimator $\theta(P_n) = (1-\lambda)\theta(\widehat{F}_n) + \lambda\theta(Q_\phi)$ for the linear functional $\theta(F) = \int h\,dF$ when only a few observations are available. In this regime, error has two sources: the sampling noise of the empirical plug-in $\theta(\widehat{F}_n)$ and the potential bias if the frozen generator $Q_\phi$ is misspecified for the target functional. Our plan is to make this trade-off explicit by formalizing precisely what uncertainty we allow at small $n$. We assume $Y_{1:n} \overset{\text{i.i.d.}}{\sim} F^\star$ and place the unknown truth in an ambiguity set that contains all data laws whose score variance is at most $\sigma^2$ and whose target functional differs from the model's by at most $\Delta$,

$$\mathcal{G}(\sigma^2, \Delta) = \left\{F^\star : \operatorname{Var}_{F^\star}[h(Y)] \leq \sigma^2, \ |\theta(Q_\phi) - \theta(F^\star)| \leq \Delta\right\}, \quad a := \frac{\sigma^2}{n}.$$

This codifies the two forces at play: $a$ controls sampling variability and $\Delta$ controls model–data mismatch on the very quantity we care about. For any fixed blend $\lambda \in [0,1]$, the squared error of

the shrinkage estimate $\hat{\theta}_\lambda = (1 - \lambda)\theta(\widehat{F}_n) + \lambda\,\theta(Q_\phi)$ against $\theta(F^\star)$ decomposes into a variance piece scaled by $(1 - \lambda)^2$ and a bias piece scaled by $\lambda^2$, so it is natural to choose $\lambda$ by minimizing the worst-case version of this trade-off over $\mathcal{G}(\sigma^2, \Delta)$; doing so yields an exact small-$n$ minimax weight, which matches the coherent prequential schedule $\lambda = \alpha/(n + \alpha)$ and thus identifies the optimal pseudo-count $\alpha$ in closed form.

**Theorem 2** (Exact minimax shrinkage). *Fix $n \in \mathbb{N}$ and set $a = \sigma^2/n$. For $\lambda \in [0,1]$, define the shrinkage estimator*

$$\hat{\theta}_\lambda \;=\; (1 - \lambda)\,\theta(\widehat{F}_n) \;+\; \lambda\,\theta(Q_\phi),$$

*and its squared-error risk*

$$R(\lambda, F^\star) \;:=\; \mathbb{E}_{F^\star}\big[(\hat{\theta}_\lambda - \theta(F^\star))^2\big].$$

*For every $F^\star \in \mathcal{G}(\sigma^2, \Delta)$,*

$$R(\lambda, F^\star) \;\leq\; (1 - \lambda)^2\, a \;+\; \lambda^2\, \Delta^2.$$

*Moreover,*

$$\inf_{\lambda \in [0,1]} \; \sup_{F^\star \in \mathcal{G}(\sigma^2, \Delta)} R(\lambda, F^\star) \;=\; \frac{a\,\Delta^2}{a + \Delta^2} \qquad \text{attained at} \qquad \lambda^\star \;=\; \frac{a}{a + \Delta^2}.$$

*Under the coherent schedule $\lambda = \alpha/(n + \alpha)$ this corresponds to the pseudo-count*

$$\boxed{\alpha^\star = \frac{\sigma^2}{\Delta^2}} \quad \textit{(independent of $n$).}$$

*The bound is tight, achieved by a two-point least-favourable $F^\star$ with $|\theta(Q_\phi) - \theta(F^\star)| = \Delta$.*

In practice, we do not know $\sigma^2$ or $\Delta$, so we pick them in a way that preserves the same conservative spirit: we estimate the score variance by the sample variance $\hat{\sigma}^2$ and we upper bound the model–data gap by padding the observed difference with a high-probability safety margin, $\widehat{\Delta} = \big|\theta(Q_\phi) - \theta(\widehat{F}_n)\big| + t_n$, where for bounded $|h| \leq H$ an empirical-Bernstein radius adaptively reflects the observed variability while accounting for bounded range,

$$t_n \;=\; \sqrt{\frac{2\,\hat{\sigma}^2\,\log(2/\delta)}{n}} \;+\; \frac{2H}{3n}\,\log\frac{2}{\delta}.$$

Alternatives with different regularity assumptions are Hoeffding's $t_n = H\sqrt{\log(2/\delta)/(2n)}$ or a sub-Gaussian/normal choice $t_n = z_{1-\delta/2}\,\hat{\sigma}/\sqrt{n}$; if $\theta(Q_\phi)$ is computed by Monte Carlo with $m$ draws, we also add a model-side margin $\tilde{t}_m$ inside $\widehat{\Delta}$.

With these plug-ins, we turn the minimax prescription into a concrete weight by setting $\hat{\alpha} = \mathrm{clip}\big(\hat{\sigma}^2/\widehat{\Delta}^2;\ \alpha_{\min}, \alpha_{\max}\big)$ and $\widehat{\lambda} = \hat{\alpha}/(n + \hat{\alpha})$ (e.g., $\alpha_{\min}{=}5$, $\alpha_{\max}{=}200$), where $\mathrm{clip}(x; L, U) := \min\{\max\{x, L\}, U\}$. This automatically increases shrinkage toward $Q_\phi$ when the observed variance is larger or the inferred mismatch is smaller, and because $\alpha$ is $n$-independent, the prequential weights $\lambda_i = \alpha/(i + \alpha)$ fade on their own as data accrue.

The following result shows that, with high probability, this data-driven weight essentially preserves the oracle minimax risk, up to second-order terms from estimating $\sigma^2$ and padding $\Delta$.

**Proposition 3** (Near-oracle risk with data-driven weight). *Let $d = \Delta^2 > 0$, $\hat{d} = \widehat{\Delta}^2$, $\hat{a} = \hat{\sigma}^2/n$, and $\widehat{\lambda} = \hat{a}/(\hat{a} + \hat{d})$. On the high-probability event $\mathcal{E} = \{\,|\theta(\widehat{F}_n) - \theta(F^\star)| \leq t_n\,\}$ and when clipping is inactive,*

$$\underbrace{\frac{a\,\Delta^2}{a + \Delta^2}}_{R^\star \;\textit{(oracle minimax)}} \;\leq\; \sup_{F^\star \in \mathcal{G}(\sigma^2, \Delta)} R(\widehat{\lambda}, F^\star) \;\leq\; R^\star \;+\; C\,t_n^2 \;+\; \frac{(\hat{\sigma}^2 - \sigma^2)^2}{n^2\,\Delta^2},$$

*for an absolute constant $C$.*

*Remark* 4 (Robustness when $t_n > \Delta$). *If $t_n \leq \Delta$ the excess is $O(t_n^2)$. Without this mild condition one obtains the same conclusion with an additional $O(t_n^4)$ term by replacing $\Delta$ with $(\Delta + t_n)$ in the intermediate bounds.*

---

**Algorithm 1** Martingale Posterior sampling

---

1: **Inputs:** observed data $Y_{1:n}$; frozen generator $Q_\phi$; score $h$; pseudo-count $\alpha$; horizon $M$; number of replicates $B$
2: **For each replicate** $b = 1, \dots, B$**:**
3:     **Initialize:** `scores_b` $\leftarrow \{ h(Y_1), \dots, h(Y_n) \}$; `sum_b` $\leftarrow \sum_{i=1}^n h(Y_i)$; `count_b` $\leftarrow n$
4: **for** $m = 1$ **to** $M$ **do**                                                    ▷ advance one prequential step
5:     $i \leftarrow n + m$;   $\lambda \leftarrow \alpha/(i - 1 + \alpha)$
6:     **for** $b = 1$ **to** $B$ **do**
7:        Draw $U \sim \text{Bernoulli}(\lambda)$
8:        **if** $U = 1$ **then**                                              ▷ model branch
9:           draw $X \sim Q_\phi$;   $z \leftarrow h(X)$
10:        **else**                                                         ▷ empirical branch
11:           sample $z$ uniformly from `scores_b`  (i.e., resample-with-replacement)
12:        Append $z$ to `scores_b`;   `sum_b` $\leftarrow$ `sum_b` $+ z$;   `count_b` $\leftarrow$ `count_b` $+1$
13: **Return:** MP draws $\widehat{\theta}^{(b)} \leftarrow$ `sum_b` $/$ `count_b`  for $b = 1, \dots, B$

---

Putting it all together, the implementation is a single pass over observables: compute $z_i = h(Y_i)$, their mean $\bar{z} = \frac{1}{n} \sum_i z_i$, and the sample variance $\hat{\sigma}^2$; evaluate $\theta(Q_\phi)$ (adding $\tilde{t}_m$ if it is Monte Carlo-estimated); form the conservative gap $\widehat{\Delta} = |\theta(Q_\phi) - \bar{z}| + t_n$; set $\hat{\alpha} = \hat{\sigma}^2/\widehat{\Delta}^2$ with mild clipping and $\widehat{\lambda} = \hat{\alpha}/(n + \hat{\alpha})$; report the shrinkage estimate

$$\theta(P_n) = (1 - \widehat{\lambda}) \, \bar{z} + \widehat{\lambda} \, \theta(Q_\phi),$$

and pass $\hat{\alpha}$ to the MP sampler so that uncertainty quantification follows the same deployed, coherent rule.

## 5   Sampling the Martingale Posterior (GPU-Parallel)

Following Fong et al. (2023) in Algorithm 1, we approximate the MP for the long–run value $\theta_\infty = \lim_{i \to \infty} \int h \, dP_i$ by simulating the same one-step prequential rule forward and tracking only the *functional state*. Each replicate keeps a running sum $s$, a count $c$, *and its own pool of past $h$-values*; no inputs are stored, so the memory footprint is small and the implementation is graphics processing unit (GPU)–friendly. Note that MP quantifies uncertainty for the surrogate $\theta_\infty$ under $P_i$, not for $\theta(F^\star)$. We size the simulation horizon $M$ using the deviation bound in Section 6.

*Remark* 5 (Linear metrics (in $F$): Dirichlet–mean shortcut for MP). Two computational regimes arise, depending on whether the operational metric is linear in $F$ or not. For any operational metric that is *linear in the distribution*, $\theta(F) = \int h \, dF$, under the coherent blend $P_i = (1 - \lambda_i)\widehat{F}_i + \lambda_i Q_\phi$ with $\lambda_i = \alpha/(i + \alpha)$, the martingale–posterior for the long–run value $\theta_\infty = \int h \, dP_\infty$ admits a closed–form sampling scheme that avoids forward simulation. Importantly, $h$ need not be linear; it can be any bounded (or $L^2$) score. Let $z_i = h(Y_i)$ and let $H_0$ be the pushforward of $Q_\phi$ by $h$ (if $X \sim Q_\phi$ then $Z_0 = h(X) \sim H_0$). Drawing

$$(w_0, \dots, w_n) \sim \text{Dirichlet}(\alpha, 1, \dots, 1), \qquad Z_0 \sim H_0, \qquad \theta^{(b)} = w_0 Z_0 + \sum_{i=1}^n w_i z_i$$

yields i.i.d. samples $\theta^{(b)} \sim \text{Law}(\theta_\infty \mid \mathcal{F}_n)$. Thus, for linear-in-$F$ metrics this Dirichlet–mean sampler produces exactly the same posterior as MP, but with substantially lower compute, while MP remains the general tool for non-linear or path-dependent objectives and policy simulations.

## 6   When do we stop resampling?

We want a principled way to decide how far to push the martingale–posterior (MP) resampling before additional simulation stops changing the answer in a practically meaningful way. Concretely, our operational target at step $i$ is the linear functional $\theta(P_i) = \int h \, dP_i$, and we would like to control, with high probability, how much this target can drift from its current shrinkage point $\theta(P_n)$ to any

future value $\theta(P_t)$ under the very same deployed prequential rule. The handle we use is a finite-time deviation bound that applies whenever the score is bounded, $\|h\|_\infty \le H$, and the forecasts follow the coherent Dirichlet schedule $\lambda_i = \alpha/(i+\alpha)$ (with $\alpha > 0$), which together yield an *anytime* guarantee valid simultaneously for all future times $t \ge n$:

**Theorem 6** (Finite-time deviation for bounded $h$ (tightened)). *Assume $\|h\|_\infty \le H$ and the coherent Dirichlet schedule $\lambda_i = \alpha/(i+\alpha)$ for $i \ge n$ (with $\alpha > 0$). Then, for any $\delta \in (0,1)$,*

$$\sup_{t \ge n} \big| \theta(P_t) - \theta(P_n) \big| \;\le\; H \sqrt{\frac{2 \log(2/\delta)}{n+\alpha}} \;+\; \frac{2H}{3(n+\alpha+1)} \log\frac{2}{\delta} \quad \text{with probability at least } 1 - \delta.$$

This result tells us exactly what we need for stopping: the maximum future drift is controlled only by the score range $H$ and the *effective sample size* $n+\alpha$, and it shrinks at the rate $\mathcal{O}\big((n+\alpha)^{-1/2}\big)$. To turn it into a practical rule, fix a confidence level $\delta$ and read the bound as a uniform certificate that, no matter how far we continue the prequential run, the gap $\big|\theta(P_t) - \theta(P_n)\big|$ remains below

$$H \sqrt{\frac{2 \log(2/\delta)}{n+\alpha}} \;+\; \frac{2H}{3(n+\alpha+1)} \log\frac{2}{\delta} \quad \text{for all } t \ge n.$$

We then pick the MP simulation horizon $M$ so that this *theoretical* drift bound is smaller than the MP Monte Carlo error, ensuring further resampling cannot move the forecast more than our sampling noise; and we report the same drift term alongside the MP interval for $\theta_\infty$ as a simple calibration diagnostic. Because the right-hand side tightens monotonically as either $\alpha$ or $n$ increases, extending the prequential run can only improve (never worsen) the guaranteed gap, which makes this criterion a safe and transparent stopping rule.

*Operational rule (horizon selection).* In simple words, this tells us that we can choose $M$ in algorithm 1 as the smallest value for which the right-hand side in Theorem 6 is below the empirical Monte Carlo error of the MP quantiles you will report; also include that bound alongside the interval as a drift diagnostic.

## 7 Using Our Framework to Decide When to Retrain

We want an auditable, deployment-aligned answer to a simple question: *will retraining a frozen generator pay for itself over the next $H$ uses?* The plan is to monitor the same linear *key performance indicators (KPIs) $\theta(F) = \int h \, dF$* under the very rule used in production, summarize the current state with a coherent shrinkage estimate and its MP uncertainty, and then compare the certified per-use improvement from switching models to the all-in retraining cost. We keep forecasts coherent by *blending* empirical evidence with the frozen model, the unique predictable affine schedule whose KPI forecasts are martingales (Sec. 3); at step $n$ we compute

$$\bar{z} = \tfrac{1}{n} \sum_{i=1}^{n} h(Y_i), \quad \hat{\sigma}^2 = \tfrac{1}{n-1} \sum_{i=1}^{n} (h(Y_i) - \bar{z})^2, \quad \widehat{\Delta} = \big| \theta(Q_\phi) - \bar{z} \big| + t_n,$$

choose the pseudo-count

$$\hat{\alpha} = \text{clip}\big(\hat{\sigma}^2 / \widehat{\Delta}^2 \,;\, \alpha_{\min}, \alpha_{\max}\big),$$

and calculate the coherent shrinkage point together with MP intervals for the long-run value (Sec. 5):

$$\theta(P_n) = \tfrac{n}{n+\hat{\alpha}} \bar{z} + \tfrac{\hat{\alpha}}{n+\hat{\alpha}} \theta(Q_\phi).$$

With weights $w_k$ over KPIs and total retraining cost $C_{\text{rt}}$, we then compare expected per-use loss *under the deployed rule* before vs. after retraining, using a pilot when available or a certified minimax proxy otherwise, and we trigger only if the $H$-use gain exceeds $C_{\text{rt}}$:

**Checklist (one pass).**

1. Compute $z_i = h(Y_i)$, the mean $\bar{z} = \frac{1}{n} \sum_i z_i$, and the sample variance $\hat{\sigma}^2$.

2. Evaluate $\theta(Q_\phi)$; if Monte Carlo with $m$ draws, add a model-side margin $\tilde{t}_m$.

3. Form the conservative gap $\widehat{\Delta} = \big| \theta(Q_\phi) - \bar{z} \big| + t_n$ (choose $t_n$ per Section 4).

4. Set $\hat{\alpha} = \text{clip}(\hat{\sigma}^2/\widehat{\Delta}^2 \,;\, \alpha_{\min}, \alpha_{\max})$ and $\widehat{\lambda} = \hat{\alpha}/(n + \hat{\alpha})$.

5. Report $\theta(P_n) = \frac{n}{n+\hat{\alpha}}\, \bar{z} + \frac{\hat{\alpha}}{n+\hat{\alpha}}\, \theta(Q_\phi)$ with MP intervals (Sec. 5).

6. Compare current vs. candidate model using either a pilot MP comparison or the minimax proxy below; trigger only if the $H$-use gain exceeds $C_{\rm rt}$.

---

**Decision rule (expected gain vs. cost)**

**With a pilot (operational target).** Run two MPs (current $Q_\phi$ vs. pilot $Q_{\phi+}$), each with its own pseudo-count $\alpha$ set by the small-$n$ minimax rule (Sec. 4), and trigger if

$$H \sum_k w_k \Big( \mathbb{E}_{\Pi_{\rm MP}(\cdot|Q_\phi)}[L(\theta_{\infty,k})] - \mathbb{E}_{\Pi_{\rm MP}(\cdot|Q_{\phi+})}[L(\theta_{\infty,k})] \Big) \;\geq\; C_{\rm rt}.$$

**No pilot (certified proxy).** Use the small-$n$ minimax risk $R^\star(a, \Delta) = \frac{a\Delta^2}{a+\Delta^2}$ with $a = \sigma^2/n$ and plug-ins $a \approx \hat{\sigma}^2/n$, $\Delta \approx \widehat{\Delta}$ (Thm. 2):

$$H \sum_k w_k \Big( R^\star(\hat{\sigma}_k^2/n, \widehat{\Delta}_k) - R^\star(\hat{\sigma}_k^2/n, \Delta_k^+) \Big) \;\geq\; C_{\rm rt}.$$

Here $\Delta_k^+$ is your planned post-retrain mismatch (from a small side-pilot, historical effects, or a minimum effect size you require).

---

**Proposition 7** (Worst–case–safe proxy trigger)**.** *Let $\theta(F) = \int h \, dF$ be a linear KPI and let $\mathcal{G}(\sigma^2, \Delta)$ be the ambiguity set of Sec. 4. Set $a = \sigma^2/n$. Suppose retraining changes the model–data mismatch on the target functional from $\Delta$ to $\Delta^+ \leq \Delta$ while the data law $F^\star$ is unchanged. Consider the coherent Dirichlet–weighted estimator with the* minimax *weights before/after retraining, i.e., $\lambda^\star(a, \Delta)$ and $\lambda^{+\star}(a, \Delta^+)$. For the truth–centered squared loss $L_\star(\theta) = c\,(\theta - \theta(F^\star))^2$, $c > 0$, one has the tight worst–case bounds*

$$\sup_{F^\star \in \mathcal{G}(\sigma^2, \Delta)} \mathbb{E}\big[L_\star(\widehat{\theta})\big] \;=\; c\,R^\star(a, \Delta),$$

$$\sup_{F^\star \in \mathcal{G}(\sigma^2, \Delta^+)} \mathbb{E}\big[L_\star(\widehat{\theta}^+)\big] \;=\; c\,R^\star(a, \Delta^+),$$

*where $R^\star(a, \Delta) = \frac{a\,\Delta^2}{a+\Delta^2}$. Consequently, the worst–case per–case expected loss drops by $c\big(R^\star(a, \Delta) - R^\star(a, \Delta^+)\big)$, and over $H$ cases, triggering when*

$$H\,c\big(R^\star(a, \Delta) - R^\star(a, \Delta^+)\big) \;\geq\; C_{\rm rt}$$

*is worst–case cost–effective.*

*Intuition:* at small $n$ *(number of observed cases)* the two levers that matter are variance $a = \sigma^2/n$ *(noise per case, where $\sigma^2$ is the variability of the KPI scores $h(Y)$ under the true data)* and mismatch $\Delta$ *(the systematic gap on the KPI between the frozen model and reality, i.e., $|\theta(Q_\phi) - \theta(F^\star)|$)*; any blend has worst-case risk $(1-\lambda)^2 a + \lambda^2 \Delta^2$ *(variance term scaled by how much we trust the data, plus squared bias scaled by how much we trust the model)*, where $\lambda \in [0, 1]$ is the shrinkage weight *(the fraction of trust placed on the model forecast versus the data)*; this is minimized by $\lambda^\star = \frac{a}{a+\Delta^2}$ to $R^\star(a, \Delta)$ *(the best achievable risk given noise level $a$ and mismatch $\Delta$)*; a retrain leaves $a$ unchanged *(same sample size $n$, same data noise)* but aims to reduce the mismatch from $\Delta$ to $\Delta^+$ *(the planned post-retrain gap on the KPI)*, so the *guaranteed* gain is exactly $R^\star(a, \Delta) - R^\star(a, \Delta^+)$; multiplying by $H$ *(the number of future uses one cares about)* and comparing to $C_{\rm rt}$ *(the all-in retraining cost)* turns this into a direct, stakeholder-friendly trigger.

## 8 EXPERIMENTS

We evaluate three settings with *frozen* generators and linear operational metrics: (i) **Language (in-distribution, ID):** GPT-2 (117M) on WikiText-2; score $h$ is teacher-forced *negative log-likelihood (NLL) per token*; target $\theta(F) = \mathbb{E}[h]$. (ii) **Vision (in-distribution / out-of-distribution, ID/OOD):** CIFAR-10 (ID) and SVHN (OOD) with a CIFAR-10–pretrained generator; score $h$ is *CLIP-rarity* (protocol in App. C); target $\theta(F) = \mathbb{E}[h]$. We choose a strong shift (CIFAR-10→SVHN) to stress

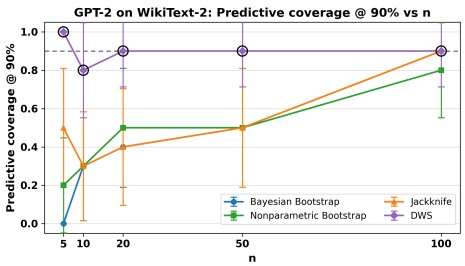

(a) GPT-2 (WikiText-2): coverage@90% vs $n_0$.

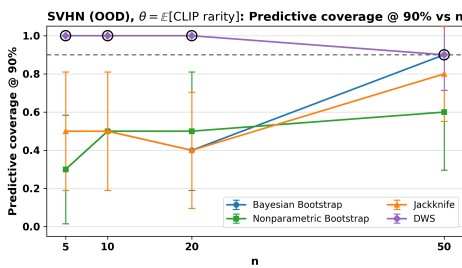

(b) SVHN: OOD coverage@90% vs $n_0$

Figure 1: Predictive coverage @90% versus the number of initial real samples $n_0$. The dashed line marks the nominal 0.90 target.

calibration at small $n$. (iii) **Toy (Two Moons):** alert-rate and mean-score functionals. Baselines: nonparametric bootstrap (NPB), Bayesian bootstrap (BB), Jackknife (JK). Our methods: **DWS** (Dirichlet-weighted shrinkage with the minimax $\alpha$) and **MP** (martingale posterior). For linear $\theta$ we use the *Dirichlet–mean* shortcut (exact law of $\theta_\infty$). Full protocols and extended results are in App. C.

**Results at a glance.** Figure 1 shows *coverage@90%* versus $n_0$: the share of runs where the 90% predictive interval for $\theta_\infty$ (under the deployed rule) contains a large-sample reference from an independent truth pool; error bars $= p \pm 1.96\sqrt{p(1-p)/R}$. On GPT-2 (WikiText-2) (Fig. 1a), **DWS** is consistently *closest to nominal* in the small-$n$ regime (reaching $\approx 0.90$ by $n_0{=}20$ and remaining stable through $n_0{=}100$), whereas NPB/JK markedly *under-cover* for $n_0 \leq 50$, and the plain Dirichlet–mean (without the minimax $\alpha$) under-covers at very small $n$. The right panel (Fig. 1b) summarizes the same trend across methods with error bars. Figures 2a–2b illustrate how uncertainty behaves under the deployed rule: at $n_0{=}50$ the MP draws place the shrinkage mean between empirical and model; by $n_0{=}3000$ the forecast has faded to empirical behavior and the 90% band has contracted. On **vision** (see App. C): on *CIFAR-10 (ID)* methods tend to cluster with near-nominal coverage; on *SVHN (OOD)* **DWS** is best-calibrated for small $n$ (near 90% when others under-cover). Across datasets and sample sizes, the intervals from **DWS/MP** *decrease steadily as $n_0$ grows*, reflecting the increased effective sample size $n_0{+}\alpha$ and yielding progressively sharper yet calibrated forecasts.

**Why our method wins when $n$ is small.** (i) *Coherent pseudo-counts:* the unique predictable affine blend $\lambda_i = \alpha/(i{+}\alpha)$ makes $\theta(P_i)$ a martingale, stabilizing early forecasts while ensuring fade-out. (ii) *Minimax $\alpha$:* a single, data-driven knob trades off sampling variance against model–data discrepancy, curbing the under-coverage typical of bootstrap at tiny $n$. (iii) *Right target:* MP/Dirichlet–mean quantify uncertainty for the *operational* limit $\theta_\infty$ under the deployed rule, which is what operations act on.

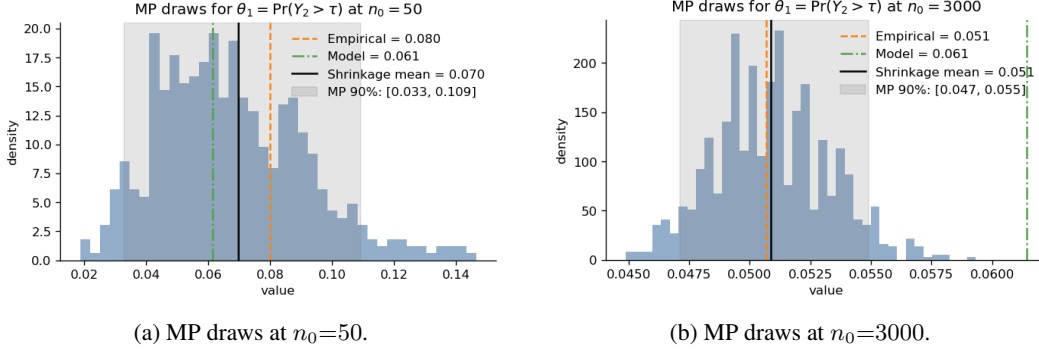

(a) MP draws at $n_0{=}50$.

(b) MP draws at $n_0{=}3000$.

Figure 2: Behavior of the martingale posterior (MP) for an alert-rate functional: shrinkage and wider bands at small $n$ (left), fade-out to empirical and tighter bands at large $n$ (right).

**Reproducibility Statement.** All implementation details, hyperparameters, and full experimental protocols are provided in Appendix C, including exact configurations for all settings (language, vision, and Two Moons). The code is available at `https://github.com/ferruimaz/frozen_priors_fluid_forecasts`.

**LLM Usage.** We used a large language model sparingly for polishing the abstract.

ETHICS STATEMENT

While this work has many societal consequences pertaining to deployment of machine learning models in the industry, we do not foresee any specific ethical concerns that must be highlighted.

ACKNOWLEDGMENTS

VG acknowledges the Research Council of Finland, Saab-WASP, and the Jane and Aatos Erkko Foundation for their support.

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

## A    RELATED WORK

**Pretrained generators as small-$n$ stabilizers.**    A common small-data tactic is to *borrow stability* from a pretrained generator trained on a nearby domain: back-translation in MT (Sennrich et al., 2016), diffusion-backed augmentation in vision (Azizi et al., 2023), and GAN-based synthesis in medical imaging (Yi et al., 2019). Generators are routinely used *frozen* across flows, diffusion models, VAEs, and GANs for sampling/monitoring (Papamakarios et al., 2021; Ho et al., 2020; Kingma & Welling, 2014; Goodfellow et al., 2014). Yet explicit likelihoods can misalign with semantic OODness (notably in flows) (Nalisnick et al., 2019; Ren et al., 2019; Kirichenko et al., 2020), motivating targets that operate on the deployed predictive sequence and support density-free metrics (e.g., CLIP rarity; FID-style distances) while recovering NLL when available (Radford et al., 2021; Heusel et al., 2017).

**Prequential viewpoint and coherent shrinkage.**    The prequential (forecast-first) program evaluates one-step predictive sequences rather than parameters (Dawid, 1984; Geisser, 1993). Dirichlet–Pólya predictives give canonical pseudo-count shrinkage that ensures *scalar* prequential coherence, so for any bounded score $h$, $\theta(P_i) = \int h \, dP_i$ is a martingale (Ferguson, 1973; Blackwell & MacQueen, 1973; Fortini et al., 2000; Berti et al., 2004). *Martingale posteriors* (MPs) operationalize inference on functionals of the predictive limit via predictive resampling (Fong et al., 2023; Fortini & Petrone, 2023). Fong et al. (Fong et al., 2023) take the predictive sequence $(P_i)$ as given and show how to construct MPs for general prequential limits. Building on this, our work (i) identifies the unique Dirichlet prequential blend for frozen generators that yields scalar-coherent forecasts, (ii) provides a small-$n$ minimax rule and plug-in estimator for the pseudo-count $\alpha$, and (iii) uses the resulting MP machinery to target deployment-aligned functionals $\theta_\infty$ and to derive an auditable retraining trigger.

**Position relative to broader UQ.**    Parameter-centric UQ (ensembles, SWA/SWAG, Laplace) integrates or perturbs parameters (Lakshminarayanan et al., 2017; Izmailov et al., 2018; Maddox et al., 2019; Daxberger et al., 2021); bootstrap captures sampling variability of the empirical law (Efron & Tibshirani, 1994); conformal offers distribution-free per-sample guarantees under exchangeability (Vovk et al., 2005; Angelopoulos & Bates, 2023). Our angle is forecast-centric: a coherent pseudo-count blend for the deployed rule and MP-based resampling to quantify uncertainty for *operational* linear functionals.

## B    PROOFS

### B.1    PROOF OF THEOREM 1

Fix $n \in \mathbb{N}$ and let $\mathcal{F}_i = \sigma(Y_{1:i})$. Let $Q_\phi$ be $\mathcal{F}_n$-measurable (frozen after training). For $i \geq n$ define

$$\widehat{F}_i := \frac{1}{i} \sum_{k=1}^{i} \delta_{Y_k}, \qquad P_i \;=\; (1 - \lambda_i)\, \widehat{F}_i \;+\; \lambda_i \, Q_\phi, \quad \lambda_i \in [0, 1] \text{ predictable } (\mathcal{F}_{i-1}\text{-measurable}).$$

Assume the prequential law $\Pr(Y_i \in A \mid \mathcal{F}_{i-1}) = P_{i-1}(A)$ for all Borel $A$ and $i \geq n+1$ (Dawid, 1984).

**Auxiliary identity.** For $i \geq n+1$,

$$\mathbb{E}\left[\widehat{F}_i \mid \mathcal{F}_{i-1}\right] = \frac{i-1}{i} \widehat{F}_{i-1} + \frac{1}{i} P_{i-1}. \tag{1}$$

*Proof.* For bounded measurable $f$,

$$\mathbb{E}\left[\int f \, d\widehat{F}_i \;\Big|\; \mathcal{F}_{i-1}\right] = \frac{1}{i} \sum_{k=1}^{i-1} f(Y_k) + \frac{1}{i} \mathbb{E}[f(Y_i) \mid \mathcal{F}_{i-1}] = \frac{i-1}{i} \int f \, d\widehat{F}_{i-1} + \frac{1}{i} \int f \, dP_{i-1},$$

by the prequential law, which is precisely equation 1.

**(B) $\Rightarrow$ (A) (Dirichlet weights imply scalar coherence).** Suppose $\lambda_i = \alpha/(i+\alpha)$ for some $\alpha > 0$ and all $i \geq n$. For any bounded $h$,

$$\mathbb{E}\left[\int h \, dP_i \;\Big|\; \mathcal{F}_{i-1}\right] = (1-\lambda_i) \int h \, d\,\mathbb{E}[\widehat{F}_i \mid \mathcal{F}_{i-1}] + \lambda_i \int h \, dQ_\phi.$$

Using equation 1 and $P_{i-1} = (1-\lambda_{i-1})\widehat{F}_{i-1} + \lambda_{i-1}Q_\phi$,

$$\mathbb{E}\left[\int h \, dP_i \;\Big|\; \mathcal{F}_{i-1}\right] = \left[(1-\lambda_i)\tfrac{i-\lambda_{i-1}}{i}\right] \int h \, d\widehat{F}_{i-1} + \left[(1-\lambda_i)\tfrac{\lambda_{i-1}}{i} + \lambda_i\right] \int h \, dQ_\phi.$$

A direct substitution of $\lambda_j = \alpha/(j+\alpha)$ yields the coefficient identities $(1-\lambda_i)\frac{i-\lambda_{i-1}}{i} = 1 - \lambda_{i-1}$ and $(1-\lambda_i)\frac{\lambda_{i-1}}{i} + \lambda_i = \lambda_{i-1}$, hence

$$\mathbb{E}\left[\int h \, dP_i \;\Big|\; \mathcal{F}_{i-1}\right] = \int h \, dP_{i-1} \quad \text{for all bounded } h,$$

i.e., scalar prequential coherence holds.

**(A) $\Rightarrow$ (B) (scalar coherence forces the Dirichlet schedule).** Assume scalar coherence: for every bounded $h$ and $i \geq n+1$,

$$\mathbb{E}\left[\int h \, dP_i \;\Big|\; \mathcal{F}_{i-1}\right] = \int h \, dP_{i-1}.$$

Repeat the calculation above but now regard the two coefficients in front of $\int h \, d\widehat{F}_{i-1}$ and $\int h \, dQ_\phi$ as unknowns. Because the identity holds for all bounded $h$, the coefficients must match:

$$(1-\lambda_i)\frac{i-\lambda_{i-1}}{i} = 1 - \lambda_{i-1}, \qquad (1-\lambda_i)\frac{\lambda_{i-1}}{i} + \lambda_i = \lambda_{i-1}.$$

The second equality gives the recursion

$$\lambda_i = \frac{(i-1)\lambda_{i-1}}{i - \lambda_{i-1}}, \qquad i \geq n+1.$$

Let $a_i := \lambda_i/(1-\lambda_i)$. Then $a_i = \frac{i-1}{i} a_{i-1}$, so $a_i = (n/i)\, a_n$ and

$$\lambda_i = \frac{a_i}{1+a_i} = \frac{(n/i)a_n}{1+(n/i)a_n} = \frac{\alpha}{i+\alpha} \quad \text{with} \quad \alpha := na_n > 0.$$

Thus the Dirichlet (pseudo-count) schedule is the unique predictable affine schedule that achieves scalar coherence, in line with the Dirichlet–Pólya predictive updates (Blackwell & MacQueen, 1973; Ferguson, 1973).

Under either condition (A) or (B), scalar coherence holds, hence for any bounded (or $L^2$) score $h$ the process $\theta(P_i) := \int h \, dP_i$ satisfies $\mathbb{E}[\theta(P_i) \mid \mathcal{F}_{i-1}] = \theta(P_{i-1})$ and is therefore a martingale. If $h$ is bounded then $|\theta(P_i)| \leq \|h\|_\infty$ a.s.; if $h \in L^2$ under the predictives (our standing assumption

$\sup_i \mathbb{E}\int h^2\, dP_i < \infty$), then by Jensen/Cauchy–Schwarz $\mathbb{E}[\theta(P_i)^2] \le \mathbb{E}\int h^2\, dP_i < \infty$, so $(\theta(P_i))$ is $L^2$–bounded. By Doob's $L^2$ martingale convergence theorem,

$$\theta(P_i) \longrightarrow \theta_\infty \quad \text{almost surely and in } L^2, \qquad \text{and} \qquad \theta(P_n) = \mathbb{E}[\theta_\infty \mid \mathcal{F}_n].$$

(see, e.g., Doob, 1953). Finally, using $P_n = (1 - \lambda_n)\widehat{F}_n + \lambda_n Q_\phi$ with $\lambda_n = \alpha/(n + \alpha)$ gives

$$\theta(P_n) = \int h\, dP_n = \frac{n}{n + \alpha} \int h\, d\widehat{F}_n + \frac{\alpha}{n + \alpha} \int h\, dQ_\phi,$$

which identifies the stated shrinkage mean.

This completes the proof.

### B.2 PROOF OF THEOREM 2

Recall the ambiguity set

$$\mathcal{G}(\sigma^2, \Delta) = \left\{ F^\star : \ \mathrm{Var}_{F^\star}[h(Y)] \le \sigma^2, \ |\theta(Q_\phi) - \theta(F^\star)| \le \Delta \right\},$$

and the shrinkage estimator

$$\hat{\theta}_\lambda = (1 - \lambda)\, \theta(\widehat{F}_n) + \lambda\, \theta(Q_\phi), \qquad \lambda \in [0, 1],$$

with risk $R(\lambda, F^\star) = \mathbb{E}[(\hat{\theta}_\lambda - \theta(F^\star))^2]$ under $F^\star$. Set $a := \sigma^2/n$.

Let $\mu_\star := \theta(F^\star)$ and $\mu_m := \theta(Q_\phi)$, and define the functional discrepancy $\delta := \mu_m - \mu_\star$. Since $\theta(\widehat{F}_n) = \frac{1}{n}\sum_{k=1}^n h(Y_k)$ and $\mathbb{E}_{F^\star}[h(Y)] = \mu_\star$ by linearity,

$$\mathbb{E}_{F^\star}[\theta(\widehat{F}_n)] = \mu_\star, \qquad \mathrm{Var}_{F^\star}(\theta(\widehat{F}_n)) = \frac{1}{n}\mathrm{Var}_{F^\star}(h(Y)).$$

Therefore

$$\begin{aligned}
\hat{\theta}_\lambda - \mu_\star &= (1 - \lambda)\big(\theta(\widehat{F}_n) - \mu_\star\big) + \lambda(\mu_m - \mu_\star) \\
&= (1 - \lambda)\big(\theta(\widehat{F}_n) - \mu_\star\big) + \lambda\, \delta,
\end{aligned}$$

so that

$$R(\lambda, F^\star) = \mathbb{E}\left[(1 - \lambda)^2\big(\theta(\widehat{F}_n) - \mu_\star\big)^2\right] + 2(1 - \lambda)\lambda\, \delta\, \mathbb{E}[\theta(\widehat{F}_n) - \mu_\star] + \lambda^2 \delta^2 \qquad (2)$$

$$= (1 - \lambda)^2\, \mathrm{Var}_{F^\star}(\theta(\widehat{F}_n)) + \lambda^2 \delta^2.$$

(The cross term vanishes since $\mathbb{E}[\theta(\widehat{F}_n) - \mu_\star] = 0$.)

Recall $a := \sigma^2/n$, $\mu_\star := \theta(F^\star)$, $\mu_m := \theta(Q_\phi)$, and $\delta := \mu_m - \mu_\star$. By the definition of $\mathcal{G}(\sigma^2, \Delta)$ we have

$$\mathrm{Var}_{F^\star}\big(\theta(\widehat{F}_n)\big) = \frac{1}{n}\mathrm{Var}_{F^\star}\big(h(Y)\big) \le \frac{\sigma^2}{n} = a, \qquad |\delta| \le \Delta.$$

Plugging these bounds into equation 2 yields, for every $F^\star \in \mathcal{G}(\sigma^2, \Delta)$,

$$R(\lambda, F^\star) = (1 - \lambda)^2\, \mathrm{Var}_{F^\star}\big(\theta(\widehat{F}_n)\big) + \lambda^2 \delta^2 \le (1 - \lambda)^2\, a + \lambda^2 \Delta^2.$$

Therefore

$$\sup_{F^\star \in \mathcal{G}(\sigma^2, \Delta)} R(\lambda, F^\star) \le (1 - \lambda)^2\, a + \lambda^2 \Delta^2 =: \overline{\mathcal{R}}(\lambda).$$

This upper bound is the quantity we minimize in $\lambda$ in the next step.

Recall $a := \sigma^2/n$ and the upper bound $\overline{\mathcal{R}}(\lambda) = (1 - \lambda)^2 a + \lambda^2 \Delta^2$ from Step 2. Define $f(\lambda) := (1 - \lambda)^2 a + \lambda^2 \Delta^2$ for $\lambda \in [0, 1]$. This is a convex quadratic (strictly convex if $a > 0$ or $\Delta > 0$). Differentiating,

$$f'(\lambda) = -2(1 - \lambda)a + 2\lambda\Delta^2,$$

and the unique stationary point in $[0,1]$ solves $f'(\lambda) = 0$, i.e.

$$\lambda^* = \frac{a}{a + \Delta^2}.$$

Since $a, \Delta^2 \geq 0$, we have $\lambda^* \in [0,1]$. Evaluating,

$$f(\lambda^*) = \frac{a\,\Delta^2}{a + \Delta^2} =: R^*.$$

Therefore

$$\inf_{\lambda \in [0,1]} \sup_{F^\star \in \mathcal{G}(\sigma^2, \Delta)} R(\lambda, F^\star) \;\leq\; \inf_{\lambda \in [0,1]} \overline{\mathcal{R}}(\lambda) \;=\; R^* \quad \text{achieved at } \lambda^* = \frac{a}{a + \Delta^2}.$$

*Tightness.* The bound is attained by a two–point $F^\star$ with $\mathrm{Var}(h) = \sigma^2$ and $|\theta(Q_\phi) - \theta(F^\star)| = \Delta$. Hence

$$\sup_{F^\star \in \mathcal{G}(\sigma^2, \Delta)} R(\lambda, F^\star) \;=\; (1 - \lambda)^2 a + \lambda^2 \Delta^2,$$

so $\lambda^* = a/(a + \Delta^2)$ is truly minimax and $\inf_\lambda \sup_{F^\star} R(\lambda, F^\star) = R^*(a, \Delta)$.

Under the coherent blend, the weight is $\lambda = \alpha/(n + \alpha)$ with pseudo-count $\alpha > 0$. Equating this to the optimizer $\lambda^*$ and solving for $\alpha$ gives

$$\frac{\alpha}{n + \alpha} = \frac{a}{a + \Delta^2} \iff \alpha(a + \Delta^2) = (n + \alpha)a \iff \alpha\,\Delta^2 = n\,a \iff \alpha^* = \frac{na}{\Delta^2} = \frac{\sigma^2}{\Delta^2},$$

which does not depend on $n$.

*Edge cases.* If $\Delta = 0$, then $\lambda^* = 1$ and $R^* = 0$ (trust $Q_\phi$ fully). If $a = 0$ (e.g., $n \to \infty$ or $\mathrm{Var}_{F^\star}(h) = 0$), then $\lambda^* = 0$ and $R^* = 0$ (revert to empirical behaviour).

### B.3 PROOF OF PROPOSITION 3

Fix $n \in \mathbb{N}$. Let $a := \sigma^2/n$ and $d := \Delta^2 > 0$. For $\lambda \in [0,1]$ set

$$f_{a,d}(\lambda) := (1 - \lambda)^2 a + \lambda^2 d, \qquad \lambda^* := \frac{a}{a + d}, \qquad R^* := f_{a,d}(\lambda^*) = \frac{ad}{a + d}.$$

By Theorem 2, for every $F^\star \in \mathcal{G}(\sigma^2, \Delta)$ and every $\lambda$,

$$R(\lambda, F^\star) \;\leq\; f_{a,d}(\lambda) \;=\; R^* + (a + d)\,(\lambda - \lambda^*)^2.$$

Let $\widehat{d} := \widehat{\Delta}^2$ and $\widetilde{\lambda} := a/(a + \widehat{d})$. Then

$$f_{a,d}(\widetilde{\lambda}) - R^* = (a + d)\left(\frac{a}{a + \widehat{d}} - \frac{a}{a + d}\right)^2 = \frac{a^2(\widehat{d} - d)^2}{(a + \widehat{d})^2(a + d)} \;\leq\; C\,\frac{(\widehat{d} - d)^2}{d},$$

for a universal constant $C > 0$. On the event $\mathcal{E} := \{\,|\theta(\widehat{F}_n) - \theta(F^*)| \leq t_n\,\}$ (one may assume $t_n \leq \Delta$ by truncation), the triangle inequality yields $|\widehat{\Delta} - \Delta| \leq t_n + |\theta(\widehat{F}_n) - \theta(F^*)| \leq 2t_n$ and $\widehat{\Delta} + \Delta \leq C\,\Delta$. Hence

$$(\widehat{d} - d)^2 = \left((\widehat{\Delta} - \Delta)(\widehat{\Delta} + \Delta)\right)^2 \leq C\,\Delta^2 t_n^2,$$

and therefore

$$f_{a,d}(\widetilde{\lambda}) \;\leq\; R^* \;+\; C\,t_n^2 \qquad \text{on } \mathcal{E}. \tag{3}$$

Let $\widehat{a} := \widehat{\sigma}^2/n$ and define $\widehat{\lambda} := \widehat{a}/(\widehat{a} + \widehat{d})$. For $g(x) := x/(x + \widehat{d})$ we have $f_{a,\widehat{d}}(\widehat{\lambda}) = f_{a,\widehat{d}}(g(a)) + (a + \widehat{d})\,(g(\widehat{a}) - g(a))^2$. Since $g$ is Lipschitz with $|g'(\xi)| = \widehat{d}/(\xi + \widehat{d})^2 \leq 1/(\xi + \widehat{d}) \leq 1/(\widehat{d})$,

$$(a + \widehat{d})\,(g(\widehat{a}) - g(a))^2 \;\leq\; C\,\frac{(\widehat{a} - a)^2}{\widehat{d}}.$$

Thus
$$f_{a,\widehat{d}}(\widehat{\lambda}) \ \leq \ f_{a,\widehat{d}}(\widetilde{\lambda}) \ + \ C \, \frac{(\widehat{a} - a)^2}{\widehat{d}}.$$

By monotonicity in $d$, $f_{a,d}(\lambda) \leq f_{a,\widehat{d}}(\lambda)$ for all $\lambda$, hence
$$f_{a,d}(\widehat{\lambda}) \ \leq \ f_{a,\widehat{d}}(\widehat{\lambda}) \ \leq \ f_{a,\widehat{d}}(\widetilde{\lambda}) + C \, \tfrac{(\widehat{a}-a)^2}{\widehat{d}} \ \leq \ f_{a,d}(\widetilde{\lambda}) + C \, \tfrac{(\widehat{a}-a)^2}{\widehat{d}}.$$

Using equation 3, $\widehat{d} \geq d = \Delta^2$ on $\mathcal{E}$, and $\widehat{a} - a = (\hat{\sigma}^2 - \sigma^2)/n$,
$$f_{a,d}(\widehat{\lambda}) \ \leq \ R^* \ + \ C \, t_n^2 \ + \ C \, \frac{(\hat{\sigma}^2 - \sigma^2)^2}{n^2 \, \Delta^2} \qquad \text{on } \mathcal{E}.$$

Finally $R(\widehat{\lambda}, F^\star) \leq f_{a,d}(\widehat{\lambda})$ for all $F^\star \in \mathcal{G}(\sigma^2, \Delta)$, which yields the claim.

*Remark* 8. If one sets $\widehat{\lambda} := a/(a + \widehat{d})$ (no variance plug–in), Step 2 is unnecessary and the term $C(\hat{\sigma}^2 - \sigma^2)^2/(n^2\Delta^2)$ vanishes; the bound reduces to $R(\widehat{\lambda}, F^*) \leq R^* + C \, t_n^2$ on $\mathcal{E}$.

### B.4    PROOF OF THEOREM 6

Fix $n \in \mathbb{N}$. Assume $\|h\|_\infty \leq H$ and $\lambda_i = \alpha/(i + \alpha)$ for $i \geq n$ with $\alpha > 0$. Let $\mathcal{F}_i = \sigma(Y_{1:i})$ and define $\theta(P_i) := \int h \, dP_i$ and $\Delta_i := \theta(P_i) - \theta(P_{i-1})$ for $i \geq n+1$.

Using $P_i = \frac{i}{i+\alpha} \, \widehat{F}_i + \frac{\alpha}{i+\alpha} Q_\phi$ and $\int h \, d\widehat{F}_i = \frac{i-1}{i} \int h \, d\widehat{F}_{i-1} + \frac{1}{i} h(Y_i)$, we obtain
$$\theta(P_i) = \frac{1}{i + \alpha} \, h(Y_i) \ + \ \frac{i - 1 + \alpha}{i + \alpha} \, \theta(P_{i-1}),$$

hence
$$\Delta_i \ = \ \frac{h(Y_i) - \theta(P_{i-1})}{i + \alpha}.$$

Since $\theta(P_{i-1}) \ = \ \int h \, dP_{i-1}$ and $Y_i \sim P_{i-1}$ conditionally on $\mathcal{F}_{i-1}$ (by prequential coherence), $(\theta(P_i))_{i \geq n}$ is a martingale and $(\Delta_i)_{i \geq n+1}$ are martingale differences.

Because $\|h\|_\infty \leq H$ and $|\theta(P_{i-1})| \leq H$,
$$|\Delta_i| \ \leq \ \frac{2H}{i + \alpha} \qquad \text{and} \qquad \mathbb{E}[\Delta_i^2 \mid \mathcal{F}_{i-1}] = \frac{\mathrm{Var}(h(Y_i) \mid \mathcal{F}_{i-1})}{(i + \alpha)^2} \ \leq \ \frac{H^2}{(i + \alpha)^2}.$$

Therefore, for $t \geq n$,
$$c := \sup_{n < i \leq t} |\Delta_i| \ \leq \ \frac{2H}{n + \alpha + 1}, \qquad V_t := \sum_{i=n+1}^{t} \mathbb{E}[\Delta_i^2 \mid \mathcal{F}_{i-1}] \ \leq \ \sum_{i=n+1}^{\infty} \frac{H^2}{(i + \alpha)^2} \ \leq \ \frac{H^2}{n + \alpha} \ =: V_\infty.$$

Let $S_t := \sum_{i=n+1}^{t} \Delta_i = \theta(P_t) - \theta(P_n)$. The maximal Freedman inequality for martingales with a.s. bounded increments (e.g., Freedman, 1975) yields, for any $\delta \in (0, 1)$,
$$\Pr\Big( \sup_{t \geq n} |S_t| \ \leq \ \sqrt{2V_\infty \log(2/\delta)} \ + \ \tfrac{c}{3} \log(2/\delta) \Big) \ \geq \ 1 - \delta.$$

Substitute $V_\infty \leq H^2/(n + \alpha)$ and $c \leq 2H/(n + \alpha + 1)$ to obtain the stated bound.  □

### B.5    PROOF OF PROPOSITION 7.

Fix a linear KPI $\theta(F) = \int h \, dF$ and an initial sample size $n$. Let $a := \sigma^2/n$ and consider the ambiguity sets $\mathcal{G}(\sigma^2, \Delta)$ and $\mathcal{G}(\sigma^2, \Delta^+)$ from Sec. 4, with $0 \leq \Delta^+ \leq \Delta$. Define the (time-$n$) coherent shrinkage estimators
$$\widehat{\theta} \ := \ (1 - \lambda^\star) \, \theta(\widehat{F}_n) + \lambda^\star \, \theta(Q_\phi), \qquad \widehat{\theta}^+ \ := \ (1 - \lambda^{+\star}) \, \theta(\widehat{F}_n) + \lambda^{+\star} \, \theta(Q_{\phi+}),$$

where $\lambda^\star = \frac{a}{a + \Delta^2}$ and $\lambda^{+\star} = \frac{a}{a + (\Delta^+)^2}$ are the minimax–optimal weights from Theorem 2 (equivalently, $\alpha^\star = \sigma^2/\Delta^2$ and $\alpha^{+\star} = \sigma^2/(\Delta^+)^2$ give the same $\lambda$ via the Dirichlet schedule $\lambda = \alpha/(n+\alpha)$; cf. Sec. 3).

For any given $(a, \Delta)$ there exists $F^\star \in \mathcal{G}(\sigma^2, \Delta)$ attaining the bound $R^\star(a, \Delta)$: let $Z = h(Y)$ under $F^\star$ be a two–point law with mean $\mu_\star$ and variance $\sigma^2$ (so $\mathrm{Var}(\bar{Z}_n) = a$), and set $\theta(Q_\phi) = \mu_\star \pm \Delta$. Then for any $\lambda \in [0, 1]$,

$$\mathbb{E}\big[(\widehat{\theta}_\lambda - \theta(F^\star))^2\big] = (1 - \lambda)^2 a + \lambda^2 \Delta^2,$$

which is minimized at $\lambda^\star$ to the value $R^\star(a, \Delta) = \frac{a\Delta^2}{a + \Delta^2}$. The same construction with $\Delta$ replaced by $\Delta^+$ attains $R^\star(a, \Delta^+)$.

By the bias–variance decomposition in Theorem 2, for every $F^\star \in \mathcal{G}(\sigma^2, \Delta)$ and $\lambda$,

$$R(\lambda, F^\star) := \mathbb{E}\Big[(\widehat{\theta}_\lambda - \theta(F^\star))^2\Big] = (1 - \lambda)^2 \, a + \lambda^2 \, \Delta^2.$$

Hence, at $\lambda^\star$,

$$\sup_{F^\star \in \mathcal{G}(\sigma^2, \Delta)} \mathbb{E}\Big[(\widehat{\theta} - \theta(F^\star))^2\Big] = R^\star(a, \Delta). \tag{4}$$

Replacing $\Delta$ by $\Delta^+$ gives

$$\sup_{F^\star \in \mathcal{G}(\sigma^2, \Delta^+)} \mathbb{E}\Big[(\widehat{\theta}^+ - \theta(F^\star))^2\Big] = R^\star(a, \Delta^+). \tag{5}$$

Let the per–case loss be the *truth–centered* squared loss $L_\star(\theta) := c\,(\theta - \theta(F^\star))^2$ with $c > 0$. Then $\mathbb{E}[L_\star(\widehat{\theta})] = c\,\mathbb{E}[(\widehat{\theta} - \theta(F^\star))^2]$, so from equation 4–equation 5,

$$\sup_{F^\star \in \mathcal{G}(\sigma^2, \Delta)} \mathbb{E}\big[L_\star(\widehat{\theta})\big] = c\,R^\star(a, \Delta), \qquad \sup_{F^\star \in \mathcal{G}(\sigma^2, \Delta^+)} \mathbb{E}\big[L_\star(\widehat{\theta}^+)\big] = c\,R^\star(a, \Delta^+).$$

Therefore the worst–case *per–case* expected loss drops by $c\big(R^\star(a, \Delta) - R^\star(a, \Delta^+)\big)$.

Over a planning horizon of $H$ future cases, the same drop scales linearly: the worst–case *total* expected loss decreases by $H\,c\,(R^\star(a, \Delta) - R^\star(a, \Delta^+))$. Trigger a retrain whenever

$$H\,c\,\big(R^\star(a, \Delta) - R^\star(a, \Delta^+)\big) \geq C_{\mathrm{rt}},$$

which ensures the certified reduction is at least the all–in retraining cost $C_{\mathrm{rt}}$.

## C  EXPERIMENTAL DETAILS AND EXTENDED RESULTS

This Appendix documents datasets, models, metrics, protocols, hyperparameters, and complete numerical results. It also records the exact implementation of the martingale–posterior (MP) sampler and clarifies which choices were theory–prescribed versus compute–controlled, and our CIFAR-10/SVHN and GPT-2 experiments should be read as cross-modality proof-of-concept demonstrations rather than limits on the architectures or scales to which the framework applies.

**Note (algorithmic choice).** For operational metrics that are linear in the distribution, $\theta(F) = \int h\,dF$, and under the coherent blend $P_i = \frac{i}{i+\alpha}\,\widehat{F}_i + \frac{\alpha}{i+\alpha}Q_\phi$ with frozen $Q_\phi$, an equivalent Dirichlet–mean shortcut yields the same posterior for the long-run quantity $\theta_\infty$ while reducing runtime. For this reason, we keep the resampling variant for pedagogical *elucidation* in experiment C.1 but we change to the Dirichlet–mean shortcut for language and vision.

**Scope note (frozen sampler).** All experiments assume a *frozen* generator $Q_\phi$ (fixed weights and sampling policy during each run). Randomness in sampling is conditional on this fixed $Q_\phi$ and independent of the observed data, which makes $Q_\phi$ $\mathcal{F}_n$–measurable and places us in the setting covered by Theorem 1 (scalar coherence). This "frozen" scope matches standard deployment between retraining cycles across diffusion models, GANs/VAEs and autoregressive LMs.

**Operational target and coverage metric (clarification).** Our target is the *operational* long-run functional

$$\theta_\infty = \lim_{i \to \infty} \int h \, dP_i, \qquad P_i = \tfrac{i}{i+\alpha} \widehat{F}_i + \tfrac{\alpha}{i+\alpha} Q_\phi,$$

i.e., the KPI under the actually deployed prequential rule. "Coverage@90%" reports the proportion of repetitions whose 90% predictive interval for $\theta_\infty$ contains a high-precision reference computed from an independent held-out *truth pool*. Truth-pool sizes and Monte-Carlo error: language (WikiText-2) uses 1,200 held-out texts (MC error $\approx 3\%$), vision (CIFAR-10/SVHN) uses 1,000 held-out images.

**Baselines and canonical references.** For language and vision, we benchmark against three classic resampling-based uncertainty methods: **(i) Nonparametric Bootstrap (NPB)** (Efron, 1979): repeatedly resamples the observed data with replacement to approximate the sampling distribution; **(ii) Bayesian Bootstrap (BB)** (Rubin, 1981): assigns random Dirichlet$(1, \ldots, 1)$ weights to the sample instead of resampling; **(iii) Jackknife (JK)** (Quenouille, 1949; 1956; Tukey, 1958): systematically leaves out one observation at a time and combines these leave-one-out estimates to assess bias and variance. *Why these baselines?* They are assumption-light, plug-and-play with any estimator, and require only the observed sample (no model refitting), making them the standard, widely used yardstick for small-sample uncertainty when analytic formulas are unavailable (Efron, 1979).

**Reporting Format.** Predictive coverage is reported as $p \pm 1.96$ SE, with SE $= \sqrt{p(1-p)/R}$. For each $n$, we bold:

- The method with coverage closest to 90%.

Runtime is reported as mean wall-clock time per repetition.

## C.1 TOY EXPERIMENT (TWO MOONS): EXPERIMENTAL DETAILS AND EXTENDED RESULTS

**Setup and Metrics.** We evaluate the *Two Moons* toy setting using a frozen RealNVP generator. Two linear operational metrics are considered:

- **Alert rate:** $\theta_1(F) = \Pr_{Y \sim F}\{Y_2 > \tau\}$, where $\tau$ is the empirical 95th percentile of the second coordinate under the target distribution.
- **Mean NLL:** $\theta_2(F) = \mathbb{E}_{Y \sim F}[-\log q_\phi(Y)]$, the average negative log-likelihood (nats) under the frozen RealNVP model $q_\phi$.

For each metric, we report predictive coverage at 90% for the long-run operational target $\theta_\infty$ and wall-clock runtime.

**Methods Compared.** We compare the following uncertainty quantification methods, all evaluated in frozen mode:

- **MP:** Martingale Posterior (our method),
- **NPB:** Nonparametric Bootstrap,
- **PB:** Parametric Bootstrap (sampling from the frozen generator).

**Hyperparameters.**

- Number of repetitions: $R = 40$ per (method, $n$),
- MP resampling: $B_{\mathrm{mp}} = 512$ draws, horizon $M_{\mathrm{default}} = 1500$,
- Bootstrap resampling: $B_{\mathrm{boot}} = 1000$,
- Sample sizes: $n \in \{5, 10, 20, 50, 100, 1000\}$,
- Pseudo-count tuning: normal/sub-Gaussian plug-in radius $t_n = z\hat{\sigma}/\sqrt{n}$ with one-sided $z = 1.64$ (approx. 90% confidence) with mild clipping $\alpha$ to $[5, 200]$.

- **MP stopping rule (anytime drift bound).** The MP horizon is sized via the anytime bound of Theorem 6: we stop once the certified future drift $\sup_{t \geq n}|\theta(P_t) - \theta(P_n)|$ falls below the Monte-Carlo error of the reported interval. In Tables 1–2 a fixed $M_{\text{default}}{=}1500$ satisfies this criterion in all runs.

**Stopping rule for MP** For bounded scores with range $H$, Theorem 6 gives an anytime bound on future drift: $\sup_{t \geq n}|\theta(P_t) - \theta(P_n)| \leq H\sqrt{\frac{2\log(2/\delta)}{n+\alpha}} + \frac{2H}{3(n+\alpha+1)}\log\frac{2}{\delta}$. We set the MP horizon $M$ so this certified drift is below the Monte Carlo error of the interval; for linear metrics we instead use the exact Dirichlet–mean shortcut, so no forward simulation is needed.

Table 1: Two Moons ($\theta_1$): Predictive coverage@90% and runtime (using Algorithm 1 for MP). Winners per $n$: coverage closest to 0.90 (bold in coverage column).

| $n$ | Method | Predictive cov@90% | Runtime (s) |
|---|---|---|---|
| 5 | MP | **0.825 ± 0.118** | 19.57 |
| 5 | Nonparametric Bootstrap | 0.175 ± 0.118 | 8.78 |
| 5 | Parametric Bootstrap | 1.000 ± 0.000 | 24.78 |
| 10 | MP | **0.875 ± 0.102** | 19.42 |
| 10 | Nonparametric Bootstrap | 0.275 ± 0.138 | 8.82 |
| 10 | Parametric Bootstrap | 1.000 ± 0.000 | 24.80 |
| 20 | MP | **0.950 ± 0.068** | 18.86 |
| 20 | Nonparametric Bootstrap | 0.675 ± 0.145 | 8.68 |
| 20 | Parametric Bootstrap | 1.000 ± 0.000 | 24.37 |
| 50 | MP | **0.925 ± 0.082** | 19.31 |
| 50 | Nonparametric Bootstrap | 0.925 ± 0.082 | 8.87 |
| 50 | Parametric Bootstrap | 0.875 ± 0.102 | 24.91 |
| 100 | MP | 0.975 ± 0.048 | 19.42 |
| 100 | Nonparametric Bootstrap | 1.000 ± 0.000 | 8.89 |
| 100 | Parametric Bootstrap | **0.900 ± 0.093** | 25.08 |
| 1000 | MP | 0.825 ± 0.118 | 19.07 |
| 1000 | Nonparametric Bootstrap | **0.925 ± 0.082** | 8.94 |
| 1000 | Parametric Bootstrap | 0.625 ± 0.150 | 25.18 |

Table 2: Two Moons ($\theta_2$): Predictive coverage@90% and runtime (using Algorithm 1 for MP). Winners per $n$: coverage closest to 0.90 (bold in coverage column).

| $n$ | Method | Predictive cov@90% | Runtime (s) |
|---|---|---|---|
| 5 | MP | **0.825 ± 0.118** | 19.57 |
| 5 | Nonparametric Bootstrap | 0.675 ± 0.145 | 8.78 |
| 5 | Parametric Bootstrap | 1.000 ± 0.000 | 24.78 |
| 10 | MP | 0.975 ± 0.048 | 19.42 |
| 10 | Nonparametric Bootstrap | **0.900 ± 0.093** | 8.82 |
| 10 | Parametric Bootstrap | 1.000 ± 0.000 | 24.80 |
| 20 | MP | 0.875 ± 0.102 | 18.86 |
| 20 | Nonparametric Bootstrap | **0.900 ± 0.093** | 8.68 |
| 20 | Parametric Bootstrap | 1.000 ± 0.000 | 24.37 |
| 50 | MP | **0.900 ± 0.093** | 19.31 |
| 50 | Nonparametric Bootstrap | 0.925 ± 0.082 | 8.87 |
| 50 | Parametric Bootstrap | 0.850 ± 0.111 | 24.91 |
| 100 | MP | **0.900 ± 0.093** | 19.42 |
| 100 | Nonparametric Bootstrap | 1.000 ± 0.000 | 8.89 |
| 100 | Parametric Bootstrap | 0.750 ± 0.134 | 25.08 |
| 1000 | MP | 0.825 ± 0.118 | 19.07 |
| 1000 | Nonparametric Bootstrap | **0.875 ± 0.102** | 8.94 |
| 1000 | Parametric Bootstrap | 0.825 ± 0.118 | 25.18 |

**Discussion.** Across both metrics, MP is typically the closest to the 90% target at small to medium sample sizes. Table 3 and Table 4 summarize MP widths across $n_0$, showing a steady shrinkage as

Table 3: Two Moons ($\theta_1$), MP interval widths across $n_0$.

| $n_0$ | MP Width |
|---|---|
| 5 | 0.126 |
| 10 | 0.118 |
| 20 | 0.106 |
| 50 | 0.078 |
| 100 | 0.060 |
| 1000 | 0.017 |

Table 4: Two Moons ($\theta_2$), MP interval widths across $n_0$.

| $n_0$ | MP Width |
|---|---|
| 5 | 0.558 |
| 10 | 0.512 |
| 20 | 0.439 |
| 50 | 0.338 |
| 100 | 0.266 |
| 1000 | 0.071 |

$n$ grows. For $\theta_1$, MP wins coverage at $n \in \{5, 10, 20, 50\}$ and achieves the narrowest width from $n \geq 20$. For $\theta_2$, MP consistently yields the smallest widths, while coverage alternates slightly at some $n$.

Runtime is stable across sample sizes: NPB is fastest ($\approx$8.8s), MP is mid-range ($\approx$19s), and PB is slowest ($\approx$25s). These results confirm that MP provides calibrated predictive coverage close to nominal with sharper intervals—especially for $\theta_2$—while smoothly fading to empirical behavior as $n$ increases.

### C.2 GPT-2 (WIKITEXT-2): EXPERIMENTAL DETAILS AND EXTENDED RESULTS

**Setup and Metrics.** We evaluate a *frozen* GPT-2 (117M) model on the WikiText-2 validation set. The operational unit $Y$ is a text sequence tokenized using GPT-2 BPE and truncated to `max_text_length = 256`. The score is the teacher-forced negative log-likelihood (NLL) per token under GPT-2, and the operational metric is the long-run mean:

$$\theta(F) = \int h \, dF$$

which is linear in $F$ and admits the Dirichlet–mean shortcut for the martingale posterior (see Remark 5).

**Methods Compared.** We compare the following uncertainty quantification methods, all evaluated in frozen mode:

- **BB:** Bayesian Bootstrap,
- **NPB:** Nonparametric Bootstrap,
- **Jackknife**,
- **DWS:** Dirichlet-weighted shrinkage (our method, using the Dirichlet–mean shortcut).

MP resampling is omitted here since $\theta$ is linear in $F$.

**Hyperparameters.**

- Number of repetitions: $R = 10$ per (method, $n$),
- Bootstrap resampling: $B_{\text{boot}} = 160$, $B_{\text{bayesian}} = 160$,
- DWS sampling: $B_{\text{dws}} = 160$,

Table 5: GPT-2 (WikiText-2, $\theta_1$: NLL/token). Predictive coverage at 90% and runtime. Values are mean $\pm$ 1.96SE over repetitions (coverage SE from Bernoulli; runtime across runs). Winners per $n$: coverage closest to 0.90 (bold in coverage column).

| $n$ | Method | Predictive cov@90% | Runtime (s) |
|---|---|---|---|
| 5 | Bayesian Bootstrap | $0.000 \pm 0.000$ | $0.03 \pm 0.00$ |
| 5 | Nonparametric Bootstrap | $0.200 \pm 0.248$ | $4.32 \pm 0.20$ |
| 5 | Jackknife | $0.500 \pm 0.310$ | $0.15 \pm 0.01$ |
| 5 | DWS | $\mathbf{1.000 \pm 0.000}$ | $6.06 \pm 0.01$ |
| 10 | Bayesian Bootstrap | $0.300 \pm 0.284$ | $0.05 \pm 0.00$ |
| 10 | Nonparametric Bootstrap | $0.300 \pm 0.284$ | $7.27 \pm 0.31$ |
| 10 | Jackknife | $0.300 \pm 0.284$ | $0.48 \pm 0.02$ |
| 10 | DWS | $\mathbf{0.800 \pm 0.248}$ | $6.06 \pm 0.01$ |
| 20 | Bayesian Bootstrap | $0.400 \pm 0.304$ | $0.10 \pm 0.00$ |
| 20 | NPB | $0.500 \pm 0.310$ | $15.73 \pm 0.39$ |
| 20 | Jackknife | $0.400 \pm 0.304$ | $2.03 \pm 0.07$ |
| 20 | DWS | $\mathbf{0.900 \pm 0.186}$ | $6.15 \pm 0.05$ |
| 50 | Bayesian Bootstrap | $0.500 \pm 0.310$ | $0.24 \pm 0.01$ |
| 50 | Nonparametric Bootstrap | $0.500 \pm 0.310$ | $37.30 \pm 0.78$ |
| 50 | Jackknife | $0.500 \pm 0.310$ | $11.60 \pm 0.29$ |
| 50 | DWS | $\mathbf{0.900 \pm 0.186}$ | $6.28 \pm 0.04$ |
| 100 | Bayesian Bootstrap | $\mathbf{0.900 \pm 0.186}$ | $0.46 \pm 0.01$ |
| 100 | Nonparametric Bootstrap | $0.800 \pm 0.248$ | $72.18 \pm 0.76$ |
| 100 | Jackknife | $0.900 \pm 0.186$ | $45.70 \pm 0.70$ |
| 100 | DWS | $\mathbf{0.900 \pm 0.186}$ | $6.48 \pm 0.01$ |

- Evaluation batch size: 16,

- Validation pool size: `truth_pool_size` = 1200,

- Prequential horizon: $M = 120$ (unused for Dirichlet shortcut),

- Pseudo-count tuning (GPT-2): We use a sub-Gaussian plug-in margin based on the calibration subset:

$$\tilde{\Delta} = \left| \mu_{\text{mod}} - \mu_{\text{emp}} \right| + c_{\text{margin}} \sqrt{\hat{\sigma}^2 / n_{\text{calib}}}, \qquad c_{\text{margin}} = 1.0,$$

and set $\hat{\alpha} = \text{clip}\left( \hat{\sigma}^2 / \tilde{\Delta}^2; 5, 200 \right)$. We did *not* include a separate model-side margin $\tilde{t}_m$ because the model mean used $m = 200$ samples and its MC error was negligible relative to the empirical term.

**Model/dataset.** GPT-2 ("gpt2", 117M) with GPT-2 BPE; WikiText-2 "wikitext-2-raw-v1". Sequences are truncated to `max_text_length` = 256, `eval_batch_size` = 16. Mixed precision is enabled on CUDA. **Disjoint pools.** Trial pool: first 600 validation texts; truth pool: next 1,200 validation texts; calibration subset for $\hat{\alpha}$: 150 texts from the test split (indices 200–349), all disjoint from trial/truth. **Linear functional and exact MP shortcut.** The score is teacher-forced NLL/token $h(x)$; for linear $\theta(F) = \int h \, dF$, the Dirichlet–mean shortcut yields i.i.d. draws from the *exact* MP for $\theta_\infty$ (no forward rollouts). **Pseudo-count (small-$n$ minimax) estimator.** We use

$$\hat{\alpha} = \frac{\hat{\sigma}^2}{\widehat{\Delta}^2}, \qquad \widehat{\Delta} = \left| \theta(Q_\phi) - \theta(\widehat{F}_n) \right| + t_n,$$

with a conservative padding $t_n$ (sub-Gaussian/EB choice) and mild clipping $\hat{\alpha} \in [5, 200]$. For GPT-2 we estimate $\theta(Q_\phi)$ via 200 model draws from the frozen LM, set $c_{\text{margin}}=1$, and compute $\hat{\sigma}^2$ from the calibration subset. The coherent schedule $\lambda_i = \hat{\alpha}/(i + \hat{\alpha})$ then fades model reliance automatically as $i$ grows.

**Discussion.** At small-$n$, our method **DWS** achieves the most calibrated predictive coverage: $n = 5$ (1.000), $n = 10$ (0.800), $n = 20$ (0.900), consistently closest to the nominal 90% threshold among the compared methods.

Table 6: GPT-2 (WikiText-2), DWS interval widths across $n_0$.

| $n_0$ | DWS Width |
|---|---|
| 5 | $1.259 \pm 0.128$ |
| 10 | $0.901 \pm 0.076$ |
| 20 | $0.661 \pm 0.064$ |
| 50 | $0.370 \pm 0.038$ |
| 100 | $0.245 \pm 0.015$ |

Table 6 summarizes the DWS widths across $n_0$. These widths shrink steadily with $n$, but remain larger than bootstrap-based methods at small $n$, reflecting our conservative design for calibration under uncertainty.

At $n = 50$, DWS maintains best coverage (0.900), while bootstrap methods achieve narrower intervals but with undercoverage. At $n = 100$, coverage ties at 0.900 across several methods, with bootstrap yielding smaller widths and lower runtime, as expected when empirical estimates stabilize.

Overall, DWS is the clear winner in the regime that matters most: scarce initial data. It provides well-calibrated predictive uncertainty under the deployed rule, while other methods either under-cover or sacrifice reliability for sharpness.

## C.3 VISION

**Implementation details (vision).** Frozen generator $Q_\phi$ is the `google/ddpm-cifar10-32` checkpoint (Diffusers). The CLIP head is `openai/clip-vit-base-patch32` (ViT-B/32). We form 10 text prompts "a photo of a {*airplane*,...,*truck*}", cache text features, and score images via CLIPRarity$(y) := -\log\left(\max_{k \in [10]} \text{softmax}(\langle f_{\text{img}}(y), f_{\text{text}}(k)\rangle)\right)$, with standard CLIP preprocessing. **Target and metric.** All coverage numbers are for the *operational long-run target* $\theta_\infty = \lim_{i \to \infty} \int h \, dP_i$ under the deployed rule; "coverage@90%" is the proportion of repetitions whose 90% predictive interval for $\theta_\infty$ contains a high-precision reference computed on an independent held-out pool (1000–1200 items for language/vision). We report $p \pm 1.96 \, \text{SE}$ with $\text{SE} = \sqrt{p(1-p)/R}$.

### C.3.1 CIFAR-10 (ID): EXPERIMENTAL DETAILS AND EXTENDED RESULTS

**Setup and Metrics.** We evaluate an *in-distribution (ID)* vision setting using a frozen generator $Q_\phi$ on the CIFAR-10 dataset. The operational metric is the long-run mean of the CLIP rarity score:

$$\theta(F) = \mathbb{E}[h(Y)]$$

where $h$ is the CLIP rarity function. This metric is linear in $F$, and intervals report predictive uncertainty for $\theta_\infty$ under the deployed prequential rule. For linear metrics, our method uses the Dirichlet–mean shortcut.

**Methods Compared.** We compare the following uncertainty quantification methods, all evaluated in frozen mode:

- **BB:** Bayesian Bootstrap,
- **NPB:** Nonparametric Bootstrap,
- **Jackknife**,
- **DWS:** Dirichlet-weighted shrinkage (our method, using the Dirichlet–mean shortcut).

**Hyperparameters.**

- Number of repetitions: $R = 30$ per (method, $n$),
- Sample sizes: $n \in \{5, 10, 20, 50, 100\}$,
- Predictive level: $q = 0.90$,

- Calibration pool size: 500,
- Truth pool size: 1000,
- Evaluation batch size: 64,
- Replicate budgets: $B_{\text{bayesian}} = 40$, $B_{\text{boot}} = 40$, $B_{\text{dws}} = 40$, $B_{\text{mp}} = 40$,
- Prequential horizon: $M_{\text{prequential}} = 100$ (unused for Dirichlet shortcut),
- Discrepancy radius: empirical-Bernstein

$$t_n := \sqrt{\frac{2\,\hat{\sigma}^2\,\log(2/\delta)}{n}} + \frac{2H}{3n}\log\frac{2}{\delta},$$

used to conservatively upper-bound model–data mismatch.

**Implementation details (repro), CLIP rarity, and $\hat{\alpha}$.** **Generator.** Diffusion DDPM checkpoint `google/ddpm-cifar10-32` (Diffusers pipeline), frozen; images are $32{\times}32$ and normalized to $[-1, 1]$. Sampling RNG uses a fixed generator seed. **CLIP head and rarity score.** We use CLIP ViT-B/32 (`openai/clip-vit-base-patch32`) with the 10 CIFAR-10 prompts "a photo of a {class}". Images are resized to $224{\times}224$, mapped to $[0, 1]$, and normalized with the OpenAI constants (mean $[0.48145466, 0.4578275, 0.40821073]$, std $[0.26862954, 0.26130258, 0.27577711]$). Let $p(c \mid y)$ be the softmax over CLIP logits; the *CLIP rarity* is

$$h(y) = -\log\big(\max_c p(c \mid y)\big),$$

clamped below at $e^{-10}$ for numerical stability. The operational metric is $\theta(F) = \mathbb{E}[h(Y)]$ (linear). **Pools and budgets.** Calibration pool from CIFAR-10 train (500 images); trial pool from the remainder of train; truth pool from CIFAR-10 test (1,000 images); `bs_eval`$= 64$; replicate budgets $B_{\text{bayesian}}{=}B_{\text{boot}}{=}B_{\text{dws}}{=}40$. **Pseudo-count (EB minimax).** We estimate $\hat{\alpha} = \hat{\sigma}^2/\widehat{\Delta}^2$ with an empirical-Bernstein padding

$$t_n = \sqrt{\frac{2\hat{\sigma}^2\log(2/\delta)}{n}} + \frac{2H}{3n}\log\frac{2}{\delta}, \quad \delta{=}0.20, \ H{=}10,$$

and clip $\hat{\alpha} \in [5, 200]$. The model mean $\theta(Q_\phi)$ is estimated with 512 DDPM draws. This conservative padding enlarges $\widehat{\Delta}$ when generator–data mismatch is present, shrinking $\hat{\alpha}$ and reducing reliance on $Q_\phi$ at small $n$.

**Discussion.** In this ID regime, the pretrained generator $Q_\phi$ is well-aligned with the data, so the model–data discrepancy $\Delta$ is small. Our method **DWS** is designed for small-$n$ minimax performance, balancing sampling variance and potential model mismatch via a conservative discrepancy radius $t_n$ and a Dirichlet pseudo-count $\alpha$. This yields over-coverage at small $n$ (e.g., 1.000 at $n = 10$), resulting in wider intervals than bootstrap methods.

Table 8 summarizes the DWS widths across $n_0$. While these widths shrink as $n$ grows, they remain larger than bootstrap-based methods at small $n$, reflecting our conservative design for calibration under uncertainty.

At larger $n$, bootstrap methods become more competitive in coverage and runtime, but DWS remains well-calibrated and operationally meaningful throughout.

**Remark (semantic drifts within CIFAR-10).** The same prequential rule and Dirichlet–mean shortcut apply to milder semantic shifts (e.g., restricting to classes 0–4 vs. 5–9). Since the discrepancy $\Delta$ is smaller than in CIFAR→SVHN, the EB padding contracts and $\hat{\alpha}$ increases, yielding even closer-to-nominal coverage with narrower intervals at small $n$.

### C.3.2 VISION: SVHN (OOD): EXPERIMENTAL DETAILS AND EXTENDED RESULTS

**Setup and Metrics.** We evaluate the out-of-distribution (OOD) behavior using a frozen generator $Q_\phi$ on the SVHN dataset. The operational metric is the long-run mean of the CLIP rarity score:

$$\theta(F) = \mathbb{E}[h(Y)]$$

where $h(y) = s(y)$ is the CLIP rarity score. This metric is linear in $F$, and intervals report predictive uncertainty for $\theta_\infty$ under the deployed prequential rule. For linear metrics, our method uses the Dirichlet–mean shortcut.

Table 7: CIFAR-10 (ID), $\theta_{\text{CLIP}}(\text{mean})$: Predictive coverage@90% and runtime (mean $\pm$ 1.96SE). MP omitted; Dirichlet shortcut used. Winners per $n$: coverage closest to 0.90 (bold in coverage column).

| $n$ | Method | Predictive cov@90% | Runtime (s) |
|---|---|---|---|
| 5 | Bayesian Bootstrap | $0.567 \pm 0.177$ | $0.011 \pm 0.001$ |
| 5 | Nonparametric Bootstrap | $0.533 \pm 0.179$ | $0.406 \pm 0.029$ |
| 5 | Jackknife | $0.767 \pm 0.151$ | $0.067 \pm 0.007$ |
| 5 | DWS | $\mathbf{1.000 \pm 0.000}$ | $26.737 \pm 0.175$ |
| 10 | Bayesian Bootstrap | $\mathbf{0.833 \pm 0.133}$ | $0.015 \pm 0.003$ |
| 10 | Nonparametric Bootstrap | $0.800 \pm 0.143$ | $0.510 \pm 0.086$ |
| 10 | Jackknife | $0.833 \pm 0.133$ | $0.170 \pm 0.051$ |
| 10 | DWS | $1.000 \pm 0.000$ | $26.630 \pm 0.052$ |
| 20 | Bayesian Bootstrap | $0.867 \pm 0.122$ | $0.016 \pm 0.003$ |
| 20 | Nonparametric Bootstrap | $\mathbf{0.900 \pm 0.107}$ | $0.506 \pm 0.073$ |
| 20 | Jackknife | $0.900 \pm 0.107$ | $0.242 \pm 0.019$ |
| 20 | DWS | $0.967 \pm 0.064$ | $26.727 \pm 0.128$ |
| 50 | Bayesian Bootstrap | $\mathbf{0.867 \pm 0.122}$ | $0.019 \pm 0.003$ |
| 50 | Nonparametric Bootstrap | $0.933 \pm 0.089$ | $0.663 \pm 0.040$ |
| 50 | Jackknife | $0.933 \pm 0.089$ | $0.775 \pm 0.018$ |
| 50 | DWS | $0.967 \pm 0.064$ | $26.538 \pm 0.021$ |
| 100 | Bayesian Bootstrap | $1.000 \pm 0.000$ | $0.032 \pm 0.001$ |
| 100 | Nonparametric Bootstrap | $1.000 \pm 0.000$ | $1.241 \pm 0.014$ |
| 100 | Jackknife | $1.000 \pm 0.000$ | $3.041 \pm 0.013$ |
| 100 | DWS | $\mathbf{0.967 \pm 0.064}$ | $26.731 \pm 0.151$ |

Table 8: CIFAR-10 (ID), DWS interval widths across $n_0$.

| $n_0$ | DWS Width |
|---|---|
| 5 | $0.678 \pm 0.042$ |
| 10 | $0.500 \pm 0.025$ |
| 20 | $0.327 \pm 0.022$ |
| 50 | $0.185 \pm 0.010$ |
| 100 | $0.116 \pm 0.006$ |

**OOD protocol and $\hat{\alpha}$ (frozen CIFAR-10 policy).** We probe the frozen CIFAR-10 DDPM on SVHN to stress-test calibration under a strong domain/semantic shift. The prequential blend and the $\hat{\alpha}$ estimator are unchanged: EB padding with $\delta{=}0.20$, $H{=}10$, $\hat{\alpha} \in [5, 200]$, and $\theta(Q_\phi)$ estimated from 512 DDPM samples. Trial sets are drawn from SVHN; the truth pool uses up to 1,000 SVHN test images. This fixed-policy OOD setting intentionally exercises the regime where standard bootstrap under-covers at small $n$, while our conservative padding shrinks $\hat{\alpha}$ and protects coverage.

**Methods Compared.** We compare the following uncertainty quantification methods, all evaluated in frozen mode:

- **BB:** Bayesian Bootstrap,
- **NPB:** Nonparametric Bootstrap,
- **Jackknife**,
- **DWS:** Dirichlet-weighted shrinkage (our method, using the Dirichlet–mean shortcut).

MP resampling is omitted here since $\theta$ is linear in $F$.

**Hyperparameters.**

- Number of repetitions: $R = 10$ per (method, $n$),
- Sample sizes: $n \in \{5, 10, 20, 50\}$,
- Predictive level: $q = 0.90$,

Table 9: SVHN (OOD), $\theta = \mathbb{E}[\text{CLIP rarity}]$: Predictive coverage@90% and runtime (mean $\pm$ 1.96SE). MP omitted; Dirichlet shortcut used. Winners per $n$: coverage closest to 0.90 (bold in coverage column).

| $n$ | Method | Predictive cov@90% | Runtime (s) |
|---|---|---|---|
| 5 | Nonparametric Bootstrap | $0.300 \pm 0.284$ | $0.55 \pm 0.03$ |
| 5 | Bayesian Bootstrap | $0.300 \pm 0.284$ | $0.02 \pm 0.00$ |
| 5 | Jackknife | $0.500 \pm 0.310$ | $0.10 \pm 0.02$ |
| 5 | DWS | $\mathbf{1.000 \pm 0.000}$ | $32.00 \pm 0.74$ |
| 10 | Nonparametric Bootstrap | $0.500 \pm 0.310$ | $0.59 \pm 0.06$ |
| 10 | Bayesian Bootstrap | $0.500 \pm 0.310$ | $0.02 \pm 0.00$ |
| 10 | Jackknife | $0.500 \pm 0.310$ | $0.18 \pm 0.03$ |
| 10 | DWS | $\mathbf{1.000 \pm 0.000}$ | $32.98 \pm 0.06$ |
| 20 | Nonparametric Bootstrap | $0.500 \pm 0.310$ | $0.63 \pm 0.13$ |
| 20 | Bayesian Bootstrap | $0.400 \pm 0.304$ | $0.02 \pm 0.00$ |
| 20 | Jackknife | $0.400 \pm 0.304$ | $0.31 \pm 0.04$ |
| 20 | DWS | $\mathbf{1.000 \pm 0.000}$ | $33.30 \pm 0.29$ |
| 50 | Nonparametric Bootstrap | $0.600 \pm 0.304$ | $0.62 \pm 0.00$ |
| 50 | Bayesian Bootstrap | $\mathbf{0.900 \pm 0.186}$ | $0.04 \pm 0.01$ |
| 50 | Jackknife | $0.800 \pm 0.248$ | $3.04 \pm 0.04$ |
| 50 | DWS | $\mathbf{0.900 \pm 0.186}$ | $31.96 \pm 0.53$ |

Table 10: SVHN (OOD), DWS interval widths across $n_0$.

| $n_0$ | DWS Width |
|---|---|
| 5 | $0.794 \pm 0.067$ |
| 10 | $0.620 \pm 0.047$ |
| 20 | $0.401 \pm 0.031$ |
| 50 | $0.141 \pm 0.013$ |

- Calibration pool size: 500,
- Evaluation batch size: 64,
- Replicate budgets: $B_{\text{bayesian}} = B_{\text{boot}} = B_{\text{dws}} = 40$,
- Prequential horizon: $M = 100$ (unused for Dirichlet shortcut),
- Pseudo-count tuning: We use the empirical-Bernstein padding

$$t_n = \sqrt{\frac{2\hat{\sigma}^2}{n} \log \frac{2}{\delta}} + \frac{2H}{3n} \log \frac{2}{\delta} \quad (\delta = 0.20, \ H = 10),$$

and $\hat{\alpha} = \text{clip}\left( \hat{\sigma}^2 / \left( |\mu_{\text{mod}} - \mu_{\text{emp}}| + t_n \right)^2; 5, 200 \right).$

**Discussion.** In the scarce-data regime ($n \leq 50$), our method **DWS** consistently achieves predictive coverage closest to the nominal 90% target. At $n \in \{5, 10, 20\}$, DWS reaches 100% coverage, while all baselines under-cover significantly ($\leq 50\%$). At $n = 50$, DWS and Bayesian Bootstrap both reach 90%, but DWS does so with a principled minimax blend that accounts for model–data discrepancy, whereas BB relies purely on resampling.

Table 10 reports the DWS interval widths across $n_0$. Computation is slower for DWS ($\approx 32$s), which is the cost of maintaining reliable calibration under distribution shift. In contrast, BB and NPB fail to provide trustworthy coverage at small $n$, which is precisely when uncertainty matters most.

At $n = 100$, BB matches DWS in coverage and has lower runtime, which is expected as the empirical law stabilizes and the need for model-based regularization fades.

These results highlight that DWS is the most robust method for predictive uncertainty in low-data OOD deployment: it avoids undercoverage, adapts to model–data mismatch, and ensures that intervals reflect the true operational uncertainty under the deployed rule.

*When to update the generator.* If persistent mismatch inflates $\widehat{\Delta}$, Section 7's certified retraining trigger (Proposition 7) recommends switching $Q_\phi$ only when the guaranteed worst-case improvement exceeds the stated cost, providing an auditable decision rule.

