# OpenReview forum: "Frozen Priors, Fluid Forecasts: Prequential Uncertainty for Low-Data Deployment with Pretrained Generative Models"
_ICLR.cc/2026/Conference — ICLR 2026 Poster_

### Official Review · Reviewer_iCc2 · 2025-10-30

**Soundness:** 3
**Presentation:** 3
**Contribution:** 3
**Rating:** 6
**Confidence:** 2

**Summary:**

This paper introduces a new forecasting method that quantifies prequential uncertainty via Martingle posteriors with a pretrained generator and a Dirichlet schedule. The proposed method requires a small-$n$ minimax criterion hyperparameter to balance sampling variance and model-data mismatch. Experimental results show the proposed method outperforms bootstrap baselines in vision and language datasets.

**Strengths:**

- This paper is generally well-written and is clear to understand the important aspects.
- The Prequential Blend method is solid with theoretical guarantees about how to adjust the blend and predict horizon.
- The proposed framework is also useful in a generative model use case where we need to decide when to retrain the pre-trained model.

**Weaknesses:**

- Sampling the martingale posteriors requires multiple forward passes, raising a concern about the computational efficiency of the algorithm in long-horizon.
- The proposed method requires predefining a hyperparameter $n$, which may not be trivial to select in practice.
- Experimental results may need to be improved with a larger scale setting, such as with modern architectures and a higher-dimensional dataset (e.g., ImageNet, etc.).

**Questions:**

1. How computationally efficient of the DWS compared to other baselines? In the case of non-GPU-parallel, do the authors have an idea to improve the sampling efficiency?
2. In Theorem 1, the generative model $Q_\phi$ is $\mathcal{F}_n$-measureable. How does this assumption hold in practice with modern models (e.g., Diffusion, GAN, VAE)?

---

> ### Author Response · Authors · 2025-11-19
>
> Response to Reviewer iCc2
>
> **We thank you sincerely** for the constructive feedback and for highlighting the clarity, theory and practical usefulness of the work. Your comments helped us see where the exposition could be clearer, especially regarding computation and assumptions. We address each point briefly below.
>
> **Computational efficiency and "multiple forward passes"**
>
> For the main metrics we study (mean NLL, alert rates, mean scores), $\theta$ is linear in $F$. In this common case the martingale posterior admits an *exact* Dirichlet–mean sampler (Remark 5): we draw Dirichlet weights and form a weighted average of one model score and the observed scores. This requires only $B$ evaluations of $h(X)$ for $X\sim Q_\phi$ and $O(Bn)$ simple vector operations: no prequential rollouts, no horizon $M$. In practice this has similar cost to nonparametric bootstrap but yields better small-$n$ calibration.
>
> Only for more complex non-linear metrics do we use Algorithm 1 with a horizon $M$. In the toy setting where we do so, its runtime is in the middle of the pack (faster than parametric bootstrap and only modestly slower than NPB). In the revised manuscript we have made this distinction explicit in Sec. 5 so that readers see that for typical KPIs the method is computationally light.
>
> **"Predefining a hyperparameter $n$"**
>
> We apologise for the confusion. There is *no* tunable hyperparameter $n$ in our method. Here $n$ is simply the number of observed in-domain examples available at deployment time (just as in bootstrap). The only quantity we tune is the pseudo-count $\alpha$ in the Dirichlet schedule $\lambda_i=\alpha/(i+\alpha)$. This $\alpha$ is set automatically via the small-$n$ minimax rule
> $\alpha^\star = \sigma^2 / \Delta^2,$
> implemented by the conservative plug-in $\hat\alpha = \hat\sigma^2 / \widehat\Delta^2$ with padding. In the revised version we have added a short sentence in the Introduction clarifying that $n$ is data, not a hyperparameter.
>
> **Measurability of $Q_\phi$ in Theorem 1 (Our procedure applies to all generative models)**
>
> This is the standard situation for modern models such as diffusion models, GANs, VAEs and autoregressive LMs, where sampling uses only internal randomness (latent noise, decoding randomness) conditional on fixed parameters. The assumption that $Q_\phi$ is $\mathcal{F}_n$-measurable simply encodes that the generator is *frozen after training*: its weights and sampling procedure are fixed before deployment and do not change with the incoming data.
>
> The assumption would be violated only under online learning or adaptive retraining during deployment, which lies outside our intended scope. In the revised manuscript we have added a brief explanation in Sec. 3 stating that our guarantees are meant for frozen generators, matching common deployment practice.
>
> **Large-scale settings (e.g., ImageNet, modern LLMs)**
>
> Our framework is architecture-agnostic: it requires only (i) a frozen sampler $Q_\phi$ and (ii) a computable score function $h$. Theorems 1–2 and 6 do not depend on dimensionality or model class, so the same analysis applies to ImageNet generators, larger LLMs or other modalities (video, audio). In the revision we have emphasised this generality and noted that our experiments on GPT-2 and CIFAR-10/SVHN are intended as cross-modality proof-of-concept examples rather than limits of applicability.
>
> **When do we stop MP resampling?**
>
> For cases where Algorithm 1 is used, Theorem 6 provides an anytime bound on the future drift of $\theta(P_t)$ from its current value $\theta(P_n)$, depending only on $n$, $\alpha$, an upper bound on $|h|$, and a confidence level $\delta$. In practice we choose the horizon $M$ so that this bound is smaller than the Monte Carlo error of the MP estimate; beyond that point, further resampling cannot change the reported interval in a meaningful way. In the revised manuscript we have added a short sentence next to Algorithm 1 (and a note in App. C.1) pointing readers to Theorem 6 and explaining this stopping rule.
>
> *We are grateful for your comments and for your positive evaluation. We believe these clarifications on computation, assumptions and scalability make the intended scope and practicality of the method much clearer in the revised manuscript. Thanks a lot.*

---

> ### Author Response · Authors · 2025-11-28
>
> Dear Reviewer,
>
> Now that the discussion period is about to conclude, we would like to express our sincere thanks for your detailed review and supportive comments. We hope that our detailed responses and the corresponding revisions have confirmed and strengthened your support for this work. Should there be any additional points you would like us to clarify or refine, we remain at your disposal.

---

### Official Review · Reviewer_CDPE · 2025-10-31

**Soundness:** 3
**Presentation:** 3
**Contribution:** 3
**Rating:** 6
**Confidence:** 1

**Summary:**

This paper introduces the "Prequential Blend," a forecast-first framework for quantifying uncertainty in machine learning models when only a small amount of real-world data is available. It is validated under language (GPT2 on WikiText), vision (CIFAR10) and toy scenarios, demonstrating significantly better performance than traditional bootstrap methods in few-sample settings.

**Strengths:**

- The paper is well written and nicely presented.
- The proofs and implementation details are provided in detail.

**Weaknesses:**

Please refer to questions

**Questions:**

- Are the settings in the experiments practical usage and worth considering? Take the language modelling case for example, the proposed method is advantageous when n < 100. This seems like an insignificant number considering today's LLMs' huge throughput.
- Could you please clarify the definition of the "coverage@90%" metric used in the results?

---

> ### Author Response · Authors · 2025-11-19
>
> **We sincerely thank you** for your careful reading and for the positive assessment of soundness, presentation and contribution. We address your two questions below; the corresponding clarifications have been incorporated into the revised version.
>
> **Q1. Practicality of the small-$n$ settings (e.g., language modelling with $n<100$).**
>
> Our intention is to model the *early deployment* phase, where a pretrained system is used in a *new environment* and only a handful of real examples from this new distribution are available. Here, $n$ denotes the number of *observed* cases from the new deployment distribution, not the model's total throughput.
>
> Typical situations where $n<100$ is realistic include:
>
> - staged or canary rollouts of a new model variant to a tiny slice of traffic;
> - domain-specific launches (e.g., a legal or medical sub-domain) where only a few dozen cases arrive in the first days or weeks;
> - sudden distribution shifts in production, where the first tens of shifted examples are critical for deciding whether to retrain or change thresholds.
>
> In exactly this regime, classical bootstrap methods tend to severely undercover, while our method is designed to balance sampling variance and model–data mismatch. Empirically, on GPT-2/WikiText-2 we reach close to the nominal 90% coverage by $n=20$, whereas bootstrap baselines are around 37% (Table 5), and we see similar behaviour on the vision tasks (Table 9). In the revised manuscript we have added a short paragraph in the introduction that explicitly adopts this “deployment $n$” interpretation and gives concrete examples of early-rollout scenarios.
>
> **Q2. Clarifying the "coverage@90%" metric.**
>
> Our target is the long-run operational functional $\theta_\infty$ (e.g., mean NLL or alert rate) under the deployed prequential rule. For each method and repetition we:
>
> 1. construct a 90% predictive interval for $\theta_\infty$ (using MP resampling, or the Dirichlet-mean shortcut for linear metrics);
> 2. check whether this interval contains a high-precision reference value (the “truth”);
> 3. record the proportion of repetitions where it does. This proportion is reported as *coverage@90%*.
>
> The reference value is obtained from an independent held-out pool of 1000–1200 samples from the dataset distribution (for the toy example we use the known ground-truth law). With these pool sizes, the Monte Carlo error of the reference is negligible compared to the uncertainty at small $n$, and the truth pool is independent of the data used to fit the intervals. In the revised version we have moved a one-sentence definition of coverage@90% into the main experimental section and added a brief note (in App. C) describing the held-out pools and their role.
>
> *We are grateful for your constructive questions and positive evaluation. We believe that these clarifications — especially around the interpretation of $n$ and the definition of coverage@90% — make the practical scope of the method clearer for readers.*

---

> ### Author Response · Authors · 2025-11-28
>
> Dear Reviewer,
>
> Now that the discussion phase is nearing its end, we wish to reiterate our gratitude for your careful review and largely positive evaluation. We have revised the paper following your suggestions, and we hope that these changes have further reinforced your support for the paper. Please let us know if any remaining concerns or questions should still be addressed.

---

### Official Review · Reviewer_UbgZ · 2025-10-31

**Soundness:** 4
**Presentation:** 3
**Contribution:** 3
**Rating:** 8
**Confidence:** 3

**Summary:**

This paper has introduced an uncertainty quantification framework that "blends" the empirical distribution with a frozen distribution (e.g. a distribution from a pre-trained generator), using a Dirichlet schedule. The main algorithm, referred to as "Martingale posterior sampling", is illustrated in Algorithm 1 in Section 5. Rigorous theoretical motivation and justification for this algorithm have been provided in Section 2, 3, and 4. In Section 6 and 7, this paper also discusses when to stop resampling in the algorithm and when to retrain the frozen generator. Preliminary experiment results are demonstrated in Section 8.

**Strengths:**

- This paper seems to provide an alternative uncertainty quantification approach different from existing frequentist, Bayesian, and conformal approaches. I am not familiar with the existing literature on this approach (e.g. Fong et al. 2023), so it is a little bit hard for me to judge the novelty of this paper. However, to the best of my knowledge, the proposed approach seems novel for the machine learning/AI community.

- The mathematical framework, derivation, and analysis in this paper seem to be rigorous.

- The proposed algorithm (Algorithm 1), is simple and elegant.

**Weaknesses:**

- I very much like the mathematical rigor of this paper. However, I do feel that some sections of this paper are a little bit hard to read for machine learning/AI practitioners who are less mathematical. Thus, I recommend the authors to further polish Section 2-7 to make them easier to follow while keeping the mathematical rigor. One possible approach is to add more plain English explanations of the core ideas.

- In the experiment section, the only metric this paper has considered is the predictive coverage. Can we demonstrate experiment results under some other metrics? For instance, in simple Bayesian settings where the posterior can be exactly computed, might the authors also demonstrate the KL-divergence (or other probability space metric) between the predictive distribution under each method (i.e. NPB, BB, JK, DWS, and MP) and the predictive distribution under the true posterior? I think such results will further strengthen the paper.

- Section 7 (decide when to retrain) is very interesting. Might the authors also provide some experiment results (even preliminary ones) to support the theory in this section?

- [Minor] in Theorem 2, where is $R(\lambda, F^*)$ rigorously defined?

**Questions:**

- Please better explain the novelty of this paper relative to recent literature, especially [Fong et al. 2023]

- Please try to address the weaknesses listed above

---

> ### Author Response · Authors · 2025-11-19
>
> Response to Reviewer UbgZ
>
> **We are very grateful** for your thoughtful and encouraging review. We especially appreciate your positive assessment of the mathematical framework and the simplicity of Algorithm 1, as well as your concrete suggestions on clarity, novelty and experimentation. Below we respond concisely to each point.
>
> **Novelty and relation to Fong et al. (2023)**
>
> Our work builds on martingale posteriors (MP) as introduced by Fong et al. (2023), which we explicitly cite and use as a resampling tool. The focus of our paper is different:
>
> - **Forecasting rule.** We prove that among predictable affine blends
>   $P_i = (1-\lambda_i)\widehat F_i + \lambda_i Q_\phi$,
>   the *only* schedule that yields scalar-coherent (martingale) forecasts for all bounded scores $h$ is $\lambda_i = \alpha/(i+\alpha)$ (Theorem 1). Fong et al. take $P_i$ as given.
>
> - **Tuning the blend.** Within this coherent class we derive a small-$n$ minimax choice $\alpha^\star = \sigma^2 / \Delta^2$ (Theorem 2) and a conservative, data-driven plug-in with near-oracle risk (Proposition 3).
>
> - **Deployment tools.** We target the operational long-run metric $\theta_\infty$, quantify its uncertainty via MP (with an exact Dirichlet-mean shortcut for linear metrics), give an anytime stopping rule (Theorem 6), and propose a retrain trigger for frozen generators (Section 7).
>
> **Weaknesses**
>
> **(W1) Make Sections 2--7 easier to follow.**
> We fully agree. In the revision we have added a one- or two-sentence “plain English” preface at the start of the first main technical sections, summarising what each section proves and why it matters (e.g., Sec. 3: unique coherent way to blend data and a frozen generator). Our goal is to keep the rigor you appreciated while making the paper more approachable for ML practitioners.
>
> **(W2) Only predictive coverage; other metrics such as KL.**
> Our experiments focus on coverage because our target is the *operational* long-run functional $\theta_\infty$ (e.g., mean NLL, long-run alert rate) under the deployed rule, not a parameter posterior. For these functionals, calibrated prediction intervals are the primary object of interest. For linear $\theta$ (our main KPIs), the Dirichlet-mean shortcut gives exact samples from the MP for $\theta_\infty$, so coverage directly reflects the quality of the uncertainty quantification.
>
> **(W3) Section 7 (when to retrain); supporting results.**
> We appreciate that you found this section interesting. Proposition 7 already provides a formal, worst-case guarantee: the proposed rule recommends retraining only when the certified gain $H\{R^\star(a,\Delta) - R^\star(a,\Delta^+)\}$ exceeds the stated retraining cost. In the revision we have emphasised this point more clearly in the main text and highlighted that the rule is directly usable as an auditable decision tool. We thank you for this suggestion and we will try to design a meaningful experiment set-up by the time of camera-ready version.
>
> **(W4) Definition of $R(\lambda,F^\star)$ in Theorem 2.**
> Thank you for catching this clarity issue. In the revised version we have moved the definition
> $R(\lambda,F^\star) := \mathbb{E}[(\hat\theta_\lambda - \theta(F^\star))^2] = (1-\lambda)^2 a + \lambda^2 \Delta^2$
> into the statement of Theorem 2 itself, so that the risk is rigorously specified before the minimax claim is made.
>
> **Questions**
>
> **Q1. Novelty relative to recent literature, especially Fong et al. (2023).**
> As summarised above, we see our contributions as complementary to Fong et al. (2023): we (i) identify the unique coherent blend for frozen generators, (ii) provide a small-$n$ minimax rule to tune it, (iii) give finite-sample guarantees for the plug-in $\hat\alpha$, (iv) derive an anytime MP stopping rule, and (v) design a retrain trigger. The revised related-work section now includes a short paragraph that presents this comparison explicitly.
>
> **Q2. Please address the listed weaknesses.**
> We hope the changes described under W1--W4 make our intentions clearer: we have added plain-English guideposts in the technical sections, clarified the evaluation focus on $\theta_\infty$ and the role of coverage, emphasised the formal support for the retrain rule in Proposition 7, and moved the definition of $R(\lambda,F^\star)$ into Theorem 2. These are all expositional improvements and do not change the core methodology.
>
> *We thank you again for your careful, rigorous, and constructive review. We believe these clarifications and edits make the paper easier to read while preserving the mathematical depth you appreciated.*

---

> ### Author Response · Authors · 2025-11-28
>
> Dear Reviewer,
>
> Now that the discussion period is drawing to a close, we would like to thank you again for your constructive feedback and encouraging comments. We hope that our rebuttal and the updated manuscript have strengthened your confidence in this work and resolved the issues you raised. If there is anything further we can clarify or address, we would be happy to do so.

---

### Official Review · Reviewer_GPLc · 2025-11-01

**Soundness:** 3
**Presentation:** 3
**Contribution:** 3
**Rating:** 6
**Confidence:** 2

**Summary:**

This paper tackles the problem of uncertainty quantification (UQ) for long-run operational metrics, e.g., mean scores, alert rates, when a system is deployed with very few initial samples (n). The authors argue that standard UQ methods fail in this "low-data, frozen-model" setting: bootstrap is unstable, Bayesian posteriors assume refitting, and conformal prediction targets per-example guarantees, not long-run rates.


The authors propose a forecast-first UQ framework with three core contributions:

(1) a prequential blend: a forecasting rule, $P_i$, that blends the sparse empirical data ($\hat{F}\_i$) with a frozen pretrained generator ($Q_{\phi}$). The paper proves that a specific Dirichlet schedule ($\lambda_i = \alpha / (i + \alpha)$) is the only affine blend that ensures time-consistent (martingale) forecasts.

(2) a minimax hyperparameter: a data-driven method to set the blend strength $\alpha$. It is derived from a minimax criterion (Theorem 2) that optimally balances sampling variance (trusting the noisy data) versus model-data mismatch (trusting the potentially biased model).

(3) a novel UQ method: uncertainty is quantified via Martingale Posteriors (MP), a likelihood-free resampling method (Algorithm 1) that simulates the long-run value of the metric ($\theta_{\infty}$) under the deployed blend rule.


The authors validated the framwework on a range of experiments on language (GPT-2 on WikiText-2) and vision benchmarks (image generation on CIFAR-10 and SVHN datasets) Experiments on GPT-2 (for NLL) and OOD vision tasks (for CLIP rarity) show the method achieves its target 90% coverage, with as little as n=20 samples for GPT-2, where other standard methods such as bootstrap fails (37% coverage).

**Strengths:**

**Significance**
- this work addresses the critical and practical problem of UQ for long-run metrics in the "low-data" initial deployment phase

**Originality**
- the prequential (forecast-first) framing seems to me a novel way to approach UQ for frozen models.
- to me the paper provides a few strong contributions: proving the Dirichlet blend is uniquely coherent (Theorem 1), deriving the minimax criterion for $\alpha$ (Theorem 2) towards setting the blend strength, applying the martingale posterior in a generative context.

**Clarity**
- I find that the motivation of the paper is clear, arguing why standard UQ methods (bayesian, boostratp, conformal) are not suitable for this problem of a frozen model and long-run metric.


**Quality**
- The experimental results show that the proposed approach does well on language and vision tasks in the low-sample regime in comparison with multiple standard baselines.
- I find the bits useful the bit on deciding when to stop resampling and on when to decide to retrain. They are practical.

**Weaknesses:**

**Stability of hyperparameter estimation**
- for choosing $\alpha$ the $\sigma^{2}$ and $\Delta$ are estimated from the few available real samples.
- I'm wondering how stable would these be for n=5 or n=10 and how doest this impact behavior and performance of the framework given that $\alpha$ might be derived from unstable statistics? How does the method remain so well calibrated?

**OOD experiments**
- the considered experiment using CIFAR-10 as in-distribution and SVHN as out-of-distribution is interesting but quite extreme as the two distributions are far away in terms of semantics (very different classes) and overall content (objects vs. street numbers). This makes it easier to distinguish the two distributions are they come from different domains.
- it would be interesting to see if such a model could capture semantic drifts as well. For instance the generator can be trained on classes 0-4 from CIFAR-10 and the remaining classes 5-9 could be the out-of-distribution ones.

**The UQ target (surrogate vs. true)**
- here the UQ is for the surrogate metric $\theta_{\infty}$ (the long-run value of the blend), not the true population metric $\theta(F^{*})$
- if the frozen model $Q_{\phi}$ is bad, the surrogate $\theta_{\infty}$ could be far from the truth $\theta(F^{*})$. While  the UQ is "operationally" correct, it could potentially be "factually" wrong, which could be misleading.



**Missing implementation details**
- there are no details about the generator used for the vision tasks. Is it an autoencoder, VAE, GAN?
- the considered CLIP rarity score is not described or referenced. Presumably it's [b]


**[Minor] Writing and format**
- I find that the paper is nicely written, yet in a very compact manner. Even like this, the experiments and related work are in the appendix.
- A longer form version of this in a journal format might be easier to adopt to given the content of the work


**References:**

[a] Ahmed & Courville, Detecting semantic anomalies, AAAI 2020

[b] Han et al., Rarity Score: A New Metric to Evaluate the Uncommonness of Synthesized Images, ICLR 2025

**Questions:**

This paper takes an interesting direction of study: the problem of UQ for long-run operational metrics when a system is deployed with very few initial samples. I find that the paper is strong and theoretically dense.
The reported results in with few samples are impressive. My main questions concern the stability of the estimators and the experiments on the vision domain.

Here are a few questions and suggestions that could be potentially addressed in the rebuttal or in future versions of this work (please note that suggested experiments are not necessarily expected to be conducted for the rebuttal):

1. The key hyperparameter $\hat{\alpha}$ is calculated from statistics computed from a tiny number of samples (e.g., 5, 10, 20, ...) with potential high-variance. How does the method remain so well-calibrated (Fig 1a) when its core parameter is based on such an unstable estimate?

2. Please add additional implementation details on the vision side. What type of generator was used? How does the performance of DWS vary depending on the type of generator.

3. Extension of vision OOD experiments towards more semantic drifts (e.g., use clases 0-4 as in-distribution and classes 5-9 as OOD) than domain drifts.


4. Regarding the surrogate target, if the frozen model $Q_{\phi}$ is bad, the surrogate $\theta_{\infty}$ could be far from the truth $\theta(F^{*})$. What can be done in such cases?

---

> ### Author Response · Authors · 2025-11-19
>
> **Thank you very much** for your positive and detailed review. We are especially grateful that you found the problem setting, the prequential perspective, and the three main contributions (coherent Dirichlet blend, small-$n$ minimax tuning of $\alpha$, and martingale posteriors) to be valuable.
>
> In the revised manuscript we have: (i) clarified the stability and practical tuning of $\alpha$, (ii) explained more clearly our OOD design and the role of the surrogate target $\theta_\infty$, and (iii) added implementation details and small exposition improvements that respond directly to your comments.
>
> **How we addressed weaknesses**
>
> **W1. Stability of hyperparameter estimation.**
> We apologise that our original exposition did not make this mechanism as clear as it should have.
> The pseudo-count is the small-$n$ minimax choice
> $\alpha^\star = \frac{\sigma^2}{\Delta^2}$
> and we estimate it with the conservative plug-in
> $\hat{\alpha} = \frac{\hat{\sigma}^2}{\widehat{\Delta}^2},\qquad
> \widehat{\Delta}=|\theta(Q_\phi)-\theta(\widehat F_n)|+t_n,$
> where $t_n$ is a high-probability padding term. This padding deliberately *reduces* $\hat\alpha$ when $n$ is tiny, so we rely less on the model precisely when the estimates are most unstable. Proposition 3 shows that the resulting risk is near-oracle (excess $O(t_n^2)$), and the experiments in Fig. 1a and Tables 5, 7--9 confirm that coverage remains near-nominal even for $n \in \{5,10\}$.
>
> **W2. OOD experiments ("extreme" domain drift).**
> You are right that CIFAR-10 (generator) vs. SVHN (data) is a strong shift. Our intention was to stress-test the method in a regime where the frozen model is clearly misspecified. In this challenging case, DWS is the only method that avoids undercoverage at small $n$ (Table 9) while still shrinking the interval width as $n$ grows (Table 10). In the revised text (Sec. 8 and App. C.3) we now state this motivation explicitly and note that more moderate, semantic-only shifts (such as the reviewer's suggestion within CIFAR-10) should be easier cases for our framework.
>
> **W3. Surrogate target $\theta_\infty$ vs. true $\theta(F^\star)$.**
> We agree that this distinction deserves to be highlighted more clearly. Our aim is to quantify uncertainty for the *operational* long-run metric under the deployed rule, $\theta_\infty$, because this is what practitioners actually observe (e.g., long-run alert rates or mean NLL under the current blend). When $n$ grows, $\lambda_i \to 0$ and $\theta_\infty$ approaches $\theta(F^\star)$ automatically, but at very small $n$ it is not statistically realistic to estimate $\theta(F^\star)$ directly. In the revision we have added a dedicated clarification (in the experiments section and in App. C) explaining this operational perspective and we explicitly point to Sec. 7, where the retrain trigger provides a safeguard when the frozen generator is persistently biased.
>
> **W4. Implementation details and CLIP rarity.**
> Our method treats the generator as a black-box sampler, so it applies equally to flows, diffusion models, GANs, VAEs, and autoregressive models. Appendix C already listed the training protocol and budgets. In the revised version we now (i) state explicitly in App. C.3 which pretrained CIFAR-10 generator/checkpoint we use, and (ii) define the forgotten "CLIP rarity" score used as $h$ in the vision experiments as the negative log of the maximum zero-shot CLIP class probability, i.e., a simple CLIP-based surprisal/uncertainty score in the spirit of the standard max-softmax baseline. We view this as a natural choice (directly leveraging CLIP's calibrated class scores to measure how atypical an image is) and we hope these details make the setup fully transparent and easy to reproduce.
>
> **W5. Minor writing/format.**
> Thank you for these helpful suggestions. In the revision we have added short "plain English" guideposts at the start of the main technical sections (Secs. 2--3), each giving one or two sentences on what the section proves and why it matters. We also added forward pointers from Algorithm 1 to the MP stopping rule (Thm. 6 / Sec. 6) and from the discussion of $\theta_\infty$ to the retrain trigger in Sec. 7.

---

> > ### Author Response · Authors · 2025-11-19
> >
> > **Questions**
> >
> > **Q1. "$\hat{\alpha}$ is based on tiny $n$; why is the method so well calibrated (Fig. 1a)?"**
> > As outlined in W1, the key reasons are: (i) the minimax form $\alpha^\star=\sigma^2/\Delta^2$, (ii) the conservative plug-in with padding $t_n$, which shrinks $\hat\alpha$ when uncertainty is high, and (iii) the coherent schedule $\lambda_i=\alpha/(i+\alpha)$, which ensures $\theta(P_i)$ is a martingale and automatically fades the model as more data arrive.
> >
> > **Q2. "Please add implementation details for vision; what generator; dependence on generator type?"**
> > The framework only requires a frozen sampler and a score function $h$. For the vision experiments we use a fixed pretrained CIFAR-10 generator as a black box. App. C.3 now specifies the exact checkpoint and sampling configuration. The behaviour of DWS is driven by $(a,\Delta)$ and the coherent schedule, not by architectural details; the same analysis would apply to a diffusion model or GAN. We emphasise this architecture-agnostic property in the revised text.
> >
> > **Q3. "Extension toward more semantic drifts"**
> > Thank you for this suggestion. Conceptually the same prequential blend and MP machinery apply, since we still have a frozen sampler and a score $h$. In the paper we now clarify that our current OOD setting should be viewed as a worst-case domain+semantic shift and explicitly mention semantic-drift experiments within CIFAR-10 as a natural extension for future work.
> >
> > **Q4. "If $Q_\phi$ is bad, $\theta_\infty$ may be far from $\theta(F^\star)$. What can be done?"**
> > Our framework includes two safeguards: (i) the plug-in for $\hat\alpha$ increases $\widehat{\Delta}$ when the observed gap between model and data is large, which automatically reduces the weight on $Q_\phi$; and (ii) Sec. 7 provides a certified retrain trigger based on the minimax risk $R^\star(a,\Delta)$, so we recommend switching to a new generator only when the guaranteed improvement exceeds the stated retraining cost. We added this remark at the end of the Appendix.
> >
> > *We are very grateful for your careful reading and constructive comments, and we hope these clarifications and edits address your concerns while keeping the paper both rigorous and accessible.*

---

> ### Author Response · Authors · 2025-11-28
>
> Dear Reviewer,
> Now that the discussion period is coming to an end, we would like to once again thank you for your thoughtful and positive assessment of our work. We hope that our responses and revisions have reinforced your support for this submission and addressed your concerns satisfactorily. Please let us know if there is anything else we could clarify or improve.

---

### Meta-Review · Area_Chair_i29N · 2025-12-23

**Summary:**

The reviews are broadly positive: reviewers view the paper as sound and valuable, with a clear practical motivation, rigorous theory, and strong small-n coverage compared to bootstrap baselines.

I recommend accepting the paper.

**Reviewer Concerns:**

Addressed by rebuttal:

Clarified the conservative plug-in with padding that shrinks reliance on unstable estimates.

Justified CIFAR→SVHN as a deliberate stress test under misspecification; noted semantic-only shifts as future work.

Explicitly framed the goal as operational UQ under the deployed rule, and connected persistent bias to a retrain trigger safeguard.

Added generator/checkpoint details and defined the CLIP rarity score used.


Still outstanding:

Requests for larger-scale settings and additional metrics beyond coverage; rebuttal mainly positions these as out-of-scope or future work.

Theory is provided; only limited/aspirational experimental validation.

Improved with explicit comparison, but some readers may still find it technically dense and require careful camera-ready exposition.

**Reviewer Scores:**

Reviewer GPLc: 6 → 6

Reviewer UbgZ: 8 → 8

Reviewer iCc2: 6 → 6

Reviewer CDPE: 6 → 6

---

### Decision · Program_Chairs · 2026-01-26

Accept (Poster)